# Ethnobotany, Phytochemistry, Biological, and Nutritional Properties of Genus *Crepis*—A Review

**DOI:** 10.3390/plants11040519

**Published:** 2022-02-14

**Authors:** Natale Badalamenti, Francesco Sottile, Maurizio Bruno

**Affiliations:** 1Department of Biological, Chemical and Pharmaceutical Sciences and Technologies (STEBICEF), Università degli Studi di Palermo, Viale delle Scienze, Ed. 17, I-90128 Palermo, Italy; natale.badalamenti@unipa.it; 2Department of Architecture, University of Palermo, Viale delle Scienze, Parco d’Orleans II, I-90128 Palermo, Italy; francesco.sottile@unipa.it; 3Centro Interdipartimentale di Ricerca “Riuszo Bio-Based Degli Scarti da Matrici Agroalimentari” (RIVIVE), Università degli Studi di Palermo, I-90128 Palermo, Italy

**Keywords:** *Crepis* ssp., Asteraceae, secondary metabolites, sesquiterpenes, Guiaianes, ethnopharmacology, biological properties

## Abstract

The genus *Crepis* L., included within the Asteraceae family, has a very wide distribution, expanding throughout the northern hemisphere, including Europe, northern Africa, and temperate Asia. This genus has a fundamental value from biodynamic and ecological perspectives, with the different species often being chosen for soil conservation, for environmental sustainability, and for their attraction towards pollinating species. Furthermore, various species of *Crepis* have been used in the popular medicine of several countries as medicinal herbs and food since ancient times. In most cases, the species is consumed either in the form of a decoction, or as a salad, and is used for its cardiovascular properties, as a digestive, for problems related to sight, for the treatment of diabetes, and for joint diseases. This literature review, the first one of the *Crepis* genus, includes publications with the word ‘*Crepis*’, and considers the single metabolites identified, characterised, and tested to evaluate their biological potential. The various isolated compounds, including in most cases sesquiterpenes and flavonoids, were obtained by extracting the roots and aerial parts of the different species. The secondary metabolites, extracted using traditional (solvent extraction, column chromatography, preparative thin layer chromatography, preparative HPLC, vacuum liquid chromatography), and modern systems such as ultrasounds, microwaves, etc., and characterised by mono- and bi- dimensional NMR experiments and by HPLC-MS, have a varied application spectrum at a biological level, with antimicrobial, antioxidant, antidiabetic, antitumor, antiviral, antiulcer, phytotoxic, and nutritional properties having been reported. Unfortunately, in vitro tests have not always been accompanied by in vivo tests, and this is the major critical aspect that emerges from the study of the scientific aspects related to this genus. Therefore, extensive investigations are necessary to evaluate the real capacity of the different species used in food, and above all to discover what the different plants that have never been analysed could offer at a scientific level.

## 1. Introduction

The genus *Crepis* L. belongs to the tribe *Cichorieae* Lam. & DC (the old name is *Lactuceae* Cass.) of the Asteraceae family. This tribe is characterised by ligulate florets, which are commonly five lobed, and by milky juice, and it includes more than 98 genera and 1550 species [1] (Figure 1). Under the tribe *Cichorieae*, there are fourteen sub-tribes, one of which is *Crepidinae*, which was re-recognised by Kilian et al. [1] with 22 genera (including *Crepis*). The genus *Crepis* L. (hawk’s beard) is the second largest genus in the tribe, with more than 200 species [2]. Actually, The Plant List [3] shows 1302 plant name records, of which 293 rank as accepted species or sub-species. Species of this genus grow in the northern hemisphere, with just a few being present in Southeast Asia. Some species also occur in East, South, and West Africa, in the Canary Islands, and Madeira. According to Babcock [4], *Crepis* originated in the Altai/Tien Shan region in Central Asia, although, at present, the centre of diversification is the circum-Mediterranean area. *Crepis* species can grow in different types of habitats ranging from alpine zone, swamps, low grasslands, and forests to beaches. Size ranges from only a few centimetres in height (e.g., *C. pygmaea*) to nearly two metres in *C. sibirica* [5].

*Crepis* L. has significant value from an agronomic and ecosystem perspective, mostly in Mediterranean environments. The role of wild weed mixtures in fields has always been complex, and now takes on renewed importance, especially today, when agriculture plays a not insignificant role in biodiversity loss [6,7]. Industrialised agricultural models, increasingly dependent on the use of synthetic chemicals (fertilisers, pesticides, and herbicides) have resulted in strong selection in wild flora with a substantial reduction in ecosystem services [8]. This specialisation is not a positive factor for either beneficial insect biodiversity [9] or improved soil characteristics and fertility preservation.

*Crepis* L. species have often been found in organically cultivated fields [10,11], i.e., fields that have not undergone invasive soil actions in which floristic communities play a key role in soil conservation and increasing ecosystem biodiversity [12]. Specifically, *Crepis sancta* is often present in natural cover crops of vineyards in France [13], contributing to the creation of planting systems with high environmental sustainability. Moreover, the presence of *Crepis* L. in wild grass mixtures appears to be significantly influenced by the fallow that preceded plantings [14]. In other environments, *C. biennis*, *C. vesicaria*, and *C. capillaris* have been used in seed mixtures to establish ecological corridors for pollinators [15,16]. *C. biennis*, moreover, has been used with positive effects in seed mixtures for vegetative cover of fallow soils in urban environments [17].

Consequently, due to our expertise in the Asteraceae family [18,19,20,21,22,23,24], and to the absence of any literature review on *Crepis* genus, we decided to analyse and survey all papers concerning this genus.

## 2. Review Methodology

In this review, a complete survey of the chemical composition, and biological properties of the essential oils, extracts, and non-volatile compounds isolated from *Crepis* taxa is provided. Moreover, the agronomical and traditional uses of *Crepis* ssp. are also reported.

The search terms included the word *Crepis*, all the botanical names of the species, both accepted names or synonyms, using the electronic databases PubMed, SciFinder, Science Direct, Scopus, Web of Science, and Google Scholar. We used the principles of PRISMA to conduct this review (Figure 2) [25]. The available information on these genera was collected from scientific databases, and cover from 1911 up to 2021. Essential oils, metabolites, traditional uses, biological activity, and toxicity are the topics of this literature review. No limitations were set for languages. Appendix A reports the taxa of *Crepis* investigated so far with their botanical authority, their synonyms, and the accepted botanical names, according to The Plant List [3].

In the present review, all the species have been reported as quoted in the original works although some of them are synonyms (Appendix A).

## 3. Alimentary Properties and Traditional Uses

### 3.1. Traditional Uses

*Crepis* species have been used as medicinal herbs and food for several centuries, and many of them are also currently in use in folk medicine. A summary of their traditional uses is presented in Table 1.

Since ancient times, in Turkish folk medicine, the decoction (5%) of the aerial parts of *C. foetida* L. subsp. *rhoeadifolia* (Bieb.) Celak, known as ‘*yürek out*’, is used for treatment of cardiovascular diseases [38,39]. Furthermore, its leaves, ‘kohum’, are consumed as food [40]. In the same area, the cooked stems of *C. sancta* (L.) Bornm. (*düğmelik*) are used as a digestive [51], whereas its raw flowers (*keklik otu*) are used for eye diseases and as vasodilators [52]. In the Van Province of Turkey, a decoction of the leaves and flowers of C. *hakkarica* Lamond, locally known as ‘*tahliş*’, is used in the treatment of diabetes [43]. The young leaves of both *C. reuteriana* Boiss., and *C. syriaca* (Bornm.) Babc. & Navashin (*souraga*) are used in Syria as salad, to relieve joint diseases pain, and as an appetiser [48].

The leaves of *C. japonica* (L.) Benth (syn. *Youngia japonica* (L.) DC.) are widely used in the traditional Chinese medicine for reducing pyrexia, detoxification, and atopy. It is antitussive, febrifuge, and also used in the treatment of boils and snakebites. Internally, it is a remedy for cold, sore throat, and diarrhoea, and externally, it is applied as a medicated paste, and relieves shingles [44,45]. In Bangladesh, the leaves of the same species, applied to wounds, act as a styptic and heals them quickly. On the other hand, the juice of the root possesses antilithic properties [46]. In China, the cough- and asthma-relieving, antipyretic and inflammation-reduction properties of *C. crocea* (Lam.) Babcock have always been known [35].

In the Himalayan region of India, the fresh juice *C. flexuosa* (DC.), where it is known as ‘*Homa Sili’*, mixed with equal amount of water, is taken regularly once a day to cure jaundice until cured [36,37].

*C. rueppellii* Sch. Bip is used both in Asia and Africa, but with different purposes. In fact, in the popular medicine of Yemen it is used for hepatic disorders (jaundice, hepatitis, and gallstones) [49], whereas in Ethiopia, the fresh roots are crushed and given orally with water to livestock against Anthrax [50]. In Africa, there are ethnopharmacological reports for another two taxa, namely *C. cameroonica* Babcock ex Hutchinson & Dalziel [syn. *C. newii* subsp. *oliveriana* (Kuntze) C.Jeffrey & Beentje] and *C. carbonaria* Sch. Bip. The first one is used for the treatment of diarrhoea, wounds, and fungal infections [30]; the second one is used for myometrial contractions [32].

In Europe, the traditional use of species of genus *Crepis* is concentrated in the Mediterranean region. In Spain, an infusion of the aerial part with flowers of *C. vesicaria* L. subsp. *haenseleri* (Boiss.) P.D. Sell [syn. *C. vesicaria* subsp. *taraxacifolia* (Thuill.) Thell.] (vernacular names *Arnica*, *Flor de arnica*) is used internally for stomach upsets, and to improve the circulation. Externally, it is used with compresses and washes, or with poultices made from the boiled plant, and is used for inflammations, as an analgesic for bruises, as cicatrizant, and as an antiseptic [59]. On the island of Crete (Greece), the leaves of *C. vesicaria* L. and *C. commutata* (Spreng.) Greuter (γλυκοσιρίδες) are usually eaten boiled in salads [33,34]. As summarised in Table 1, a body of information has been reported on the use of *Crepis* taxa in Italy. The young leaves of *C. sancta* (L.) Bornm. are greatly appreciated in Emilia Romagna (Central Italy), where it is known as ‘*ciocapia*t’, as food, pan-fried or as salad, and for its diuretic and laxative effects [53]. They are also eaten, boiled or raw, in salads and soups in Tuscany and Umbria (Central Italy) [31,54]. Other species used as food in some parts of Italy and with therapeutic properties include *C. bursifolia* L. (diuretic and antoxidant, Sicily) [27,29], *C. capillaris* (L.) Wallr., *C. leontodontoides* All. (laxative, Tuscany) [31], *C. lacera* Ten. (diuretic and hypoglycemic) [47], *C. vesicaria* L. (hypoglycaemic, laxative and hypertensive, Sardinia) [57,58] and *C. vesicaria* L. subsp. *taraxicifolia* (Thuill) Thell. (diuretic and laxative, Emilia Romagna) [53].

### 3.2. Nutritional Composition

Spontaneous green leaf plants contain important quantities of essential macro- and micro-nutrients, and their content regulates the appearance of degenerative diseases, such as cancer, or cardiovascular diseases, etc. The nutritional content of wild species from the island of Crete, *C. commutata* and *C. vesicaria*, used in the culinary field as salads, was investigated in a study. The quantity of α-tocopherol was evaluated (0.36 mg/100 g for *C. commutata*, and 0.41 mg/100 g for *C. vesicaria*), which is a fat-soluble antioxidant that intervenes in the glutathione peroxidase pathway, protecting cell membranes from oxidation, and reacting with the lipid radicals produced in the chain reaction of lipid peroxidation, along with the total content of phenolic compounds (49.08 and 49.42 mg/100 g, respectively), and the content of the main minerals such as Na, Ca, Mg, K, Fe, and nitrites. The results suggested that a diet rich in these two plants provides the daily human needs with an excellent amount of vitamins, antioxidants, and minerals [60]. In turn, the nutritional content of the leaves of *C. vesciaria* subsp. *taraxacifolia* has been studied. It emerged that carbohydrates (with maltose being the most abundant of all) represented the most abundant macronutrient, followed by ash, proteins, and lipids. The lipid profile was moderate (0.69 g/100 g of fresh plant), with a predominance of polyunsaturated fatty acids, such as α-linoleic acid. Additionally, in this case, the content of macro-minerals (Na, Ca, Fe, K, Mg, etc.) was moderate, corroborating the work performed by Simopoulos and Gopalan [60], and suggesting that the consumption of this plant as a supplementary food contributes to a balanced diet [61].

Finally, *C. vesciaria* L., normally consumed in the Mediterranean diet, has been tested for its nutritional composition and content of carotenoids, tocols, thiamine, and riboflavin. Small amounts of thiamine and riboflavin were found, but the plant was a source of xanthophylls (violaxanthin, neoxanthin, lutein, zeaxanthin and *β*-cryptoxanthin) and carotenes (*α*-carotene, *β*-carotene, 9-*cis*-*β*-carotene, and 13-*cis*-*β*-carotene). Lutein accounted for the highest content (about 4 mg/100 g), but with good amounts of tocol, in particular α-tocopherol (about 2*–*3 mg/100 g), suggesting it as a source of fibre, as well as vitamins A and E [62].

## 4. Non-Volatile Metabolites

### 4.1. Occurrence

In this section, the occurrence of secondary metabolites from roots and aerial parts of Crepis ssp. is investigated. The main class of compounds were sesquiterpenes, although several flavonoids, aromatics, and other mebolites were also identified.

The identification of single isolated metabolites was conducted by spectroscopical methods as mass spectrometry (MS) infrared spectroscopy (IR), optical rotation, and 1D- and 2D-NMR. The analysis of several extracts was carried out by HPLC-MS. The analyses of oils were performed by GC-MS.

### 4.2. Sesquiterpenes

Differently from other tribes of the Asteraceae, which contain numerous metabolites with different sesquiterpene carbon skeletons, *Cichorieae* show the presence of only eudesmanes, germacranes, and guaianes. A common feature of sesquiterpenoids from the *Cichorieae* tribe is the presence of sugar or carboxylic acid residues in the molecules. Guaianolides are the most diversified group of sesquiterpenoids within the *Cichorieae*, and according to Zidorn [63], they can be divided in 13 different classes.

Phytochemical research on the thirty-one species of *Crepis*, studied so far for their sesquiterpenoidic content (Table 2), showed that those ones present in *Crepis* taxa belong to only three classes: the Costus lactone type, by far the most represented, with fifty-two compounds (**1**–**52**), hypocretenolides (**53**–**55**), and lactucin type (**56**–**62**) (Figure 3, Figure 4, Figure 5, Figure 6, Figure 7, Figure 8 and Figure 9).

It is noteworthy that all guaianolides belonging to the Costus lactone type, grouped on the basis of their exocyclic double bonds (Figure 3, Figure 4, Figure 5 and Figure 6), have a C-6/C-12 *γ*-lactone with a *trans* junction and that, apart from 9*α*-hydroxy-3-deoxyzaluzanin C (**1**) [73], they all possess an oxygenated function at C-3. Other oxygenated functions can occur on C-8 and/or C-9, but never on C-1, C-2, C-5, and C-11. The three hypocretenolides (**53**–**55**) (Figure 7) were detected only in the roots of *C. aurea* (L.) Cass [65], whereas the presence of lactucin derivatives (Figure 8) has been observed only in aerial parts of *C. dioscoridis* L. (8-deoxylactucin, **56**) [74], *C. leontodontoides* All. [78], and *C. sancta* (L.) Bornm. [78].

A very small numer of sesquiterpenes with different skeletons—eudesmane (**63**–**69**), and germacrane (**71**–**74**) (Table 2, Figure 9)—have been detected in *Crepis* species. The former occurred in *C. pygmaea* L. [86,87], C. napifera (Franch.) Babcock [82], and *C. sancta* (L.) Bornm. [90,91], whereas germacranolide was present only in *C. mollis* (Jacq.) Asch. [80], *C. incana* Sm. [76], and *C. napifera* (Franch.) Babcock [83].

### 4.3. Flavonoids

The occurrence of flavonoids, detected in the twenty-six *Crepis* taxa studied so far, is reported in Table 3, and their structures are depicted in Figure 10.

In contrast to the large diversity of sesquiterpenes reported, only a limited number of flavonoids have been observed in the genus *Crepis*. These are mainly luteolin derivatives, with luteolin itself (**76**) being the most common one, present in twenty-one species. Additionally, luteolin-7-*O*-glucoside (**80**) and luteolin 7-*O*-glucuronide (**81**) are present in several species, eighteen and thirteen, respectively, whereas luteolin 7-*O*-gentiobioside (**82**) and luteolin 4′-*O*-glucoside (**83**) occur in few cases. Among the flavonols, the most common one is quercetin (**84**), present in *C. divaricata* Boss. & Heldr, *C. pygmaea* L. [103] and *C. foetida* L. ssp. *rhoeadifolia* (Bieb.) Celak. [104].

Quite interesting is the research carried out by Zidorn et al. [102], who investigated the influence of the altitude of the collection site on the quantitative content of flavonoids within the flowering heads of *C. capillaris* (L.) Wallr. growing in New Zealand. The results showed a clear increase in flavonoid content in relation to the altitude. Similar data were observed for taxa of genus *Leontodon* collected in Europe [109]. It was hypothesised that higher altitude sites are exposed to higher peak UV-B levels, and therefore plants synthesize and store higher levels of flavonoids as UV-B protective compounds in flowering heads.

### 4.4. Other Metabolites

Derivatives of caffeic acid (**98**) are well-known antioxidant metabolites widely represented in several wild herbs of the Asteraceae family such as arnica (Arnica montana L.), chamomiles [*Chamaemelum nobile* (L.) All.], feverfew [*Tanacetum parthenium* (L.) Sch. Bip.], giant goldenrod (*Solidago gigantea* Aiton), milfoil (*Myriophyllum* ssp.), mugwort (*Artemisia vulgaris* L.), and tansy (*Tanacetum vulgare* L.) [110]. They are also present in many of the *Crepis* species studied so far, and their occurrence and structures are shown in Table 4 and Figure 9, respectively. The main metabolites of this class are caffeic acid (**98**), chlorogenic acid (**104**), caffeoyltartaric acid (**105**), 3,5-di-*O*-dicaffeoylquinic acid (**106**), 3,4-di-*O*-caffeoylquinic acid (**107**), cichoric acid (**108**), and rosmarinic acid (**112**), found, with different occurrence, in twenty *Crepis* taxa. Other metabolites detected in *Crepis* ssp., including free carboxylic acids (**95**-**97**, **100**, **101**, **126**), lignans (**121**, **122**), eugenyl derivatives (**114**, **115**), triterpenoids (**127**-**130**), steroids (**131**, **132**), etc., are reported in Table 4 and Figure 11 and Figure 12. 

Some studies have been carried out on the composition of the fixed oils from seeds [115,116,117,118] and aerial parts [61,112] of *Crepis* ssp., and their composition is reported in Table 5. Several fixed oils are characterised by the presence of two unusual fatty acids, vernolic and crepenynic acid. Vernolic acid is used in the manufacture of plastic formulations and paints, whereas crepenynic acid is potentially a useful intermediate in the chemical synthesis and production of conjugated triene intermediates [115].

## 5. Essential Oils

Despite the large number of species of genus *Crepis*, to the best of our knowledge, only one paper has been published on the chemical composition of their essential oils. It concerns the study of the volatile components of two odoriferous *Crepis* spp., namely *C. rubra* L. and *C. foetida* L., collected in Greece [119]. The volatile constituents of *C. rubra* flowers were characterised by the presence of sterols (38.9%), fatty acids and esters (33.2%), and hydrocabons (22.5%), with *β*-sitosterol (38.9%), eicosanoic acid (30.2%) and heneicosane (17.0%) being the main constituents. On the other hand, the stems–leaves volatiles revealed the absence of sterols and the occurrence of large amount of hydrocabons (41.8%), followed by fatty acids and esters (33.2%), and diterpenes (8.6%). The main constituents of this oil were heneicosane (11.4%), phytol (8.6%), eicosanoic acid (7.4%), salicylaldehyde (6.4%), hexadecanol (5.7%), and nonanal (5.0%). The oil of the flowers of *C. foetida* L. was characterised by the presence hydrocabons (47.3%) and phenylpropanoids (40.0%), with salicylaldehyde (40.0%), heneicosane (18.2%), and octadecanol (15.9%) being the main constituents of the oil. On the other hand, principal metabolites of the stems–leaves of *C. foetida* L. was carvacrol (51.0%), although large amount of salicylaldehyde (30.1%) was also detected.

## 6. Biological Properties

In the different subsections of Section 6, the different activities performed using different ssp. of *Crepis* genus are taken into consideration.

### 6.1. Antibacterial and Antifungal Activity

Antimicrobial resistance (AMR), which has sharply increased in recent years [120], is leading to an inability to treat different infectious diseases [121]. Bacteria, fungi, viruses, protozoa can show AMR towards the action of drugs. In fact, the massive use of antimicrobial prescriptions, and the excessive use of antibiotics in the agro-food system have contributed to the rapid increase in AMR. Well-founded studies have now shown that the annual deaths worldwide caused by AMR are equal to 750,000 units, and it is assumed that they will be more than 10 million by 2050 [122].

To counter this effect, secondary metabolites [123,124,125], produced by plant species as defence mechanisms, could represent a resource to be exploited for obtaining new drugs. Additionally, synthetic compounds [126,127,128] can be used to counteract the attack of parasites and pathogens before they can cause serious structural damage.

Ndom et al. [30] demonstrated that three lactone sesquiterpenes, 3*β*,9*β*-dihydroxyguaian-4(15),10(14),11(13)-trien-6,12 olide (**2**), 3*β*,8*α*-dihydroxyguaian -4(15),10(14),11(13)-trien-6,12 olide (8-desacylcynaropicrin) (**3**), and 8*α*-hydroxy-4*α*(13),11*β*(15)-tetrahydrozaluzanin C (**47**), chromatographically isolated from the aerial parts of *C. camerronica*, possess antibacterial activity against *Staphylococcus aureus* ATCC 13709, *Escherichia coli* ATCC 25922, and *E. coli* ATCC 35218 (inhibitory zone between 5 and 16 mm), slightly lower than that of the streptomycin control (inhibitory zone between 2.5–6 mm); however, none of the three compounds was found to be active against *Pseudomonas aeruginosa* ATCC 27853, *Candida albicans* ATCC 10231, and *Klebsiella pneumoniae* ATCC 10031.

The methanolic extract obtained from *C. sancta* leaves was evaluated against six strains of *S. aureus* isolated from soccer player’s shoes in Balikesir Spor (U16 and U17) soccer team using the spread-plate method. The antibacterial results, obtained both as zone of inhibition, in mm, and by broth dilution method, indicated that *C. sancta* extract showed a weak antibacterial activity only against the *S. aureus* BFT1 culture (inhibitory zone: 10 mm; MIC = 3250 μg/mL). The antibiotics oxacillin, vancomycin, and erythromycin were used as positive controls, while methanol was used as negative control [129]. In a painstaking work [130], the antimicrobial activity of the extract obtained from the infusion of *C. foetida* L. in boiling water was evaluated against four Gram-positive microorganisms (*Streptococcus pneumoniae* ATCC 19615, *S. pyogenes* ATCC 13615, *S. aureus* ATCC 25923, and *S. epidermidis* ATCC 12228), four Gram-negative bacteria (*Klebsiella pneumoniae* RSKK 574, *Haemophilus influenzae* ATCC 49766, *P. aeruginosa* ATCC 10145, and *Acinetobacter baumannii* RSKK 02026), and four fungal strains (*C. albicans* ATCC 10231, *C. tropicalis*, *C. parapsilosis* ATCC 22019, and *C. krusei*). The MIC value obtained against each microorganism was determined using broth microdilution techniques. The results showed a moderate activity of the aforementioned extract both towards Gram-negative and Gram-positive bacteria (MIC ranging from 32 to 64 μg/mL), but excellent activity against the tested fungal strains (MIC values between 8–32 μg/mL). Interestingly, the extract (MIC = 32 μg/mL) was 2 times more active than fluconazole control (MIC = 64 μg/mL) against the *C. krusei* strain.

### 6.2. Antioxidant Activity

The so-called reactive oxygen species (ROS), such as the superoxide anion (O_2_^•−^), and hydroxyl radical (HO^•^), are commonly produced by various biochemical reactions, and can produce, if present in high concentrations, serious damage to DNA, proteins, lipids, thus causing aging and cell death [131]. Scientific research has been addressed towards the study of new antioxidant compounds, obtained both from the natural source [132,133,134,135,136] and chemical-synthetic procedures [137], able to prevent and control the treatment of oxidative stress diseases, including cancer, cardiovascular and neurodegenerative diseases. It is also known that the antioxidant activity of a plant extract is highly correlated with the presence of phenolic compounds [138], and especially of phenolic acids [139]. Following phytochemical investigations, it emerged that species of the genus *Crepis* contain a good number of phenolic acids [61,140], inhibitors of oxidative processes. The HPLC-MS analysis of the aerial parts and roots of *C. foetida* L. subsp. *rhoeadifolia* showed the presence of chlorogenic acid (**104**) and luteonin (**76**). The tested extracts showed antioxidant activity in assays of DPPH (IC_50_ = 0.26–1.54 mg/mL) and of the reactive substances of thiobarbituric acid (TBARS) (MDA level = 4.54–19.15 nmol/mL), thus suggesting that the activity shown by the extracts of this plant is linked to its polyphenolic content [46]. In two other scientific works, the infusion, obtained in boiling water from the aerial parts of *C. foetida* L. [130], and the methanolic extract of *C. sancta* [129] were evaluated for their possible antioxidant capacity. None of the in vitro tests carried out with ABTS decolourisation assay and DPPH assay showed radical scavenging activity.

Finally, the hydro-alcoholic extract of the dehydrated leaves of *C. vesicaria* subsp. *taraxacifolia*, rich in L-chicoric acid (**108**) (130.5 mg per gram of plant), was evaluated for its antioxidant activity by DPPH, ABTS, and FRAP methods. The results showed that the extract had a good radical-scavenging activity and antioxidant action (DPPH = IC_50_ 26.20 μg/mL; ABTS = IC_50_ 18.92 μg/mL; FRAP = 0.68 μg/mL of Trolox Equivalent) [61].

The absence of other assays besides the chemistry methods (DPPH, ABTS, FRAP) is a shortcoming, and further future studies in vivo would be desirable.

### 6.3. Anti-Inflammatory Activity

In the sole work conducted to evaluate the volatile content of *Crepis* species, the flowers, leaves, and stems of *C. foetida* and *C. rubra* [119] were investigated. As previously stated in Chapter 5, the main components of *C. foetida* were aromatic compounds such as salicylaldehyde (30.1–40.0%) and carvacrol (51.0%, present only in the stems), whereas the essential oils of *C. rubra* were characterised by the abundance of *β*-sitosterol (38.9%, only in flowers), eicosanoic acid (7.4–30.2%), and eneicosane (11.4–17.0%). The fraction rich in salicylaldehyde from the essential oil of *C. foetida* flowers was studied for anti-inflammatory activity in vitro at different concentrations (6.25, 12.5, 25.0, 50.0 μg/mL) in the HMEC-1 cell line for the reduction of ICAM-1 expression induced by TNF-α. The activities highlighted a powerful reduction in ICAM-1 expression at 65–70% compared to the control. Further in vivo studies are needed to evaluate the potential of this essential oil.

In another paper, the ethanolic extract of *C. vesciaria* subsp. *taraxacifolia* leaves, rich in phenolic acids, namely derivatives of caffeic and ferulic acid as well as isomers of chicoric acid, was quantified by HPLC-PDA (high-performance chromatography with a diode array detector), showing inhibition of NO production in a dose-dependent manner with an IC_50_ of 0.43 ± 0.01 mg/mL [61], confirming the use of this plant for inflammatory problems [46].

### 6.4. Antidiabetic Activity

Some *Crepis* species, due to the presence of numerous components, such as phenolic acids, steroids, and lignanoids, are used for the treatment of diabetes [43] and can significantly reduce blood glucose, increasing insulin resistance [114]. Li et al. [35] demonstrated, after having characterised the ethanolic extract obtained from *C. crocea* (Lam.) Babc., that the medium and high dose (80 mg/kg/d and 160 mg/kg/d, respectively) of three branched polysaccharides (CTP3-B, CTP3-C and CTP3-D) in KK-Ay mice resulted in a significant decrease in body weight or a reduction in the levels of blood glucose compared to the control group. This in vivo study suggests that regular use of polyacids can be employed in the treatment of type 2 diabetes and plays a key role in juggling glucose levels in the blood. Further studies should be conducted to evaluate the real mechanism of action of these acids and understand the hypoglycaemic effect.

### 6.5. Antiulcer Activity

Taraxinic acid-1′-*O*-*β*-*D*-glucopyranoside (**72**), a glycosylated sesquiterpene extracted from the roots of C. napifera, has been reported to have antiulcer action, inhibiting significantly (*p* < 0.01), at a dose of 70 mg/kg, the development of gastric lesions induced in rats by aspirin, and not influencing the release of gastric secretion stimulated by histamine in the animal’s stomach [83].

The methanolic extract of the widespread *C. sancta* [91], rich in 3-oxo-*γ*-costic acid (**68**), its methyl ester (**69**), and seven different methoxylated flavonoids (Table 3) was tested in vivo for a possible antiulcer action. At doses of 100 mg/kg and 200 mg/kg, the extract tested on male albino mice showed an excellent antiulcer activity, comparable or even better than omeprazole, a typical drug used in antigastric activity. The histopathological examination showed a clear reduction in the infiltration of inflammatory cells, and the cessation of mucosal haemorrhage, with a dose-dependent reduction in the volume and titratable acidity of gastric juice which may explain the remarkable gastroprotective effect.

### 6.6. Antiviral Activity

Viruses represent a serious global, health, and even economic threat [141]. The past two years have taught us seriously what the explosion of new viruses entails. Scientific research has investigated possible antiviral agents starting from natural secondary structures. In a 2004 work, Ooi and collaborators [106] emphasised excellent antiviral activities of *Youngia japonica* (syn. *Crepis japonica*) ethanol extract against respiratory syncytial virus (RSV) in HEp-2 cells, capable of showing an inhibitory action equal to 50% (IC_50_) in doses between 3.0 and 6.0 g/mL, comparable values obtained from the action of the drug ribavirin. The selective index (SI), which is the ratio of the maximum non-cytotoxic concentration (MNCC) at the IC_50_ of the anti-RSV activity, was between 75 and 150, confirming the activity had taken over. The ethanolic extract, chromatographically focused, led to the isolation of three polyphenolic derivatives, 3,4-dicaffeylquinic acid (**107**), 3,5-dicaffeylquinic acid (**106**), and luteolin-7-*O*-glucoside (**80**). Both caffeic acid derivatives demonstrated strong antiviral activity against RSV in HEp-2 cells with an IC_50_ of 0.5 μg/mL, while no anti-RSV activity was shown for the flavonoid [44]. Further structure–activity investigations are needed to understand the mechanism of action of caffeine derivatives.

### 6.7. Cytotoxic Activity

Eudesmane sesquiterpenes, 3-oxo-di-nor-eudesm-4-en-6*β*-hydroxy-11-oic (**66**), 3-oxo-6*β*-hydroxy-*γ*-costic acid (**67**), 3-oxo-*γ*-costic acid (**68**), 3-oxo-*γ*-costic acid methyl ester (**69**), and (6*R*,9*S*)-roseoside (**70**), and two known methylated flavonoids acid from the acetone and the methanol extracts of *C. sancta* were subjected to an in vitro cytotoxicity assay against mouse lymphoma (L5178Y) cells. Only 3-oxo-*γ*-costic acid (**68**) and its methyl ester (**69**) demonstrated moderate cytotoxic activities (IC_50_ of 21.0 and 9.5 μM), while the other three sesquiterpenes showed weak cytotoxic effects compared to kahalalide F, used as a control (IC_50_ = 4.3 μM). The antiproliferative activity of methyl ester was greater than that of the acid equivalent, with an IC_50_ value of 9.5 μM, and it can be attributed to increased lipophilicity of the methylene derivative favouring a better interaction with the target cellular receptor [90]. In another study, polar extracts obtained from Oriental hawksbeard [*Youngia japonica* (L.) DC.] were tested for potential anticancer activity in vitro against three cell lines: human promyelocytic leukemia (HL-60), human myeloid leukemia (chronic K-562), and mouse sarcoma 180 (S-180). The aqueous extract of *Youngia japonica* (syn. *C. japonica*) showed an inhibitory rate between 80 and 90% at the maximum dose of 450 g/mL on the proliferation of HL-60 and K-562 cells, but significantly less activity in the suppression of S-180 cells (inhibition = 38%) [106]. Not enough information can be gleaned from the work of Mikropoulou et al. [140]. In fact, only the decoction of *C. sancta* was cytotoxic at the dose tested against C5N and A5 using the MTT test.

Finally, the methanolic extract of *C. foetida* L. subsp. *rhoeadifolia* (Bieb.) Celak. inhibited the proliferation of HEPG-2, Caco-2, and MCF-7 cells with IC_50_ values of 119.61, 101.20, and 90.94 g/mL, respectively. No inhibition was shown towards the proliferation of MCF-10A cells [104]. The major criticality is due to the fact that they are all tests carried out in vitro, using extracts in most of the experiments. There is a need to test the individual molecules, or all of them together, evaluating the synergistic effect and the mechanism of action.

### 6.8. Toxicity

Herbal medicines are often used to control various diseases. However, the presence of natural toxins and pollutants can cause severe adverse reactions, such as severe liver damage [142]. For example, in some plants belonging to the Asteraceae family, such as *Doronicum orientale* Hoffm., there are toxins such as pyrrolizidine alkaloids which, if ingested incorrectly or in large quantities, can cause biochemical reactions with serious damage to the liver [143]. In this context, the traditional use of *C. rueppellii* Sch. Beep. in the treatment of hepatic syndromes, including jaundice and hepatitis, was confirmed by a preliminary study by Fleurentin et al. [49]. The ethanolic extract of *C. rueppellii* at a dose of 200 mg/kg showed hepatoprotective properties against hepatotoxicity in mice induced by injection of specific doses of ethanol and carbon tetrachloride, significantly reducing the mortality of the treated subjects (*p* < 0.05) and the significant increase in plasma GPT activity (*p* < 0.01). The lower and higher doses (100 mg/kg and 400 mg/kg, respectively) did not show hepatoprotective effects. The anti-necrotic action, however, must be demonstrated and evaluated through biochemical studies trying to correlate it with any specific active principles.

Three sesquiterpene lactones, 8-*epi*-desacylcynaropicrin-3-*O*-*β*-glucopyranoside (**10**), 8-epigrosheimin (**24**), and 8-*β*-hydroxidehydrozaluzanine C (**5**), isolated from the aerial parts of *C. lacera*, were examined by in vitro and in vivo studies to evaluate their relative toxicity. Indeed, several farmers in southern Italy have complained of the death of grazing ruminants, such as sheep and goats, after the ingestion of large quantities of *C. lacera*. Cell viability was evaluated in cell cultures of the Madin Darby Bovine Kidney (MDBK) bovine kidney cell line after incubation with active molecules. The three sesquiterpenes were shown to be cytotoxic after 48 h of incubation, and in vivo studies on adult male Sprague-Dawley rats showed significant lesions (*p* < 0.05) in the liver, lungs and kidneys when treated with 2 mg/kg of *C. lacera* polar extract [47]. In-depth studies need to be conducted to understand the mechanism of action of the secondary metabolites present in this plant.

## 7. Conclusions

Agricultural landscapes have changed substantially in recent decades, from the dominance of smallholder fields with wide diversity to large-scale monoculture systems where ecosystem services are much more challenged. These species, with annual or biennial behaviour, especially when in relationship and balance with other species, make a substantial contribution to the development of resilient patterns that respond to the principles of agroecology and enhance pathways of ecological transition.

In this context, *Crepis* L. has attracted great interest in many Mediterranean environments. In fact, since ancient times, the *Crepis* genus has been used in the agri-food field. Up to January 2022, 132 metabolites have been isolated from the roots and aerial parts of the different species, identified using spectroscopic techniques, and evaluated for biological potential.

Among these, the guaianic-type sesquiterpenes and the glycosylated and non-glycosylated methylated flavonoids are the most characteristic. In particular, phytochemical studies have shown the presence of typical sesquiterpene structures: the costus lactone type, by far the most represented, with fifty-two compounds, hypocretenolides, and the lactucin type. Especially in the last two decades, the extracts obtained from the different parts of the plants and the pure isolated compounds have been tested and evaluated for their biological activities, such as antitumor, anti-inflammatory, antiviral, antimicrobial, antiulcer, antioxidant, and nutritional. However, the few studies on biological properties of *Crepis* species present in literature do not make it possible to confirm and validate the edibility of this genus or its use, in the form of a decoction, in traditional medicine.

Nevertheless, major critical issues remain. Most scientific studies have aimed at testing the extracts or compounds isolated through in vitro studies; biological support obtained in vivo and biochemical investigations relating to the mechanism of action of the tested samples are lacking. Only one work reports the isolation of essential oils, which could be used for a chemotaxonomic distinction between dubious species.

This literature review aims to direct current researchers to work on a widely promising genus at the agri-food level, given the massive incidence of numerous biologically exploitable sesquiterpene compounds, due to the different structural variety.

## Figures and Tables

**Figure 1 plants-11-00519-f001:**
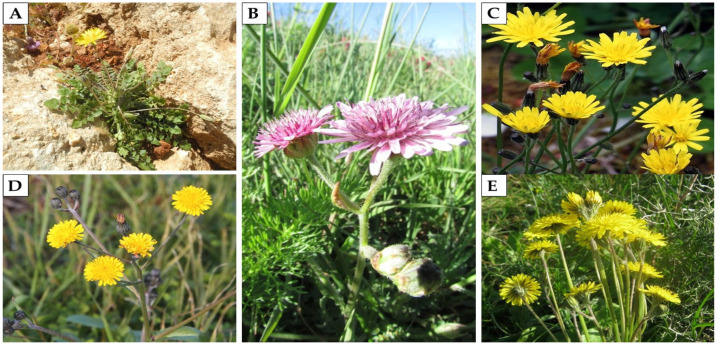
Some of the *Crepis* ssp. present in Sicily, Italy: (**A**) *C. bursifolia* L., (**B**) *C. bivoniana* (Rchb.) Soldano & F. Conti, (**C**) *C. leontodontoides* All., (**D**) *C. vesicaria* L., and (**E**) *C. sancta* (L.) Bornm.

**Figure 2 plants-11-00519-f002:**
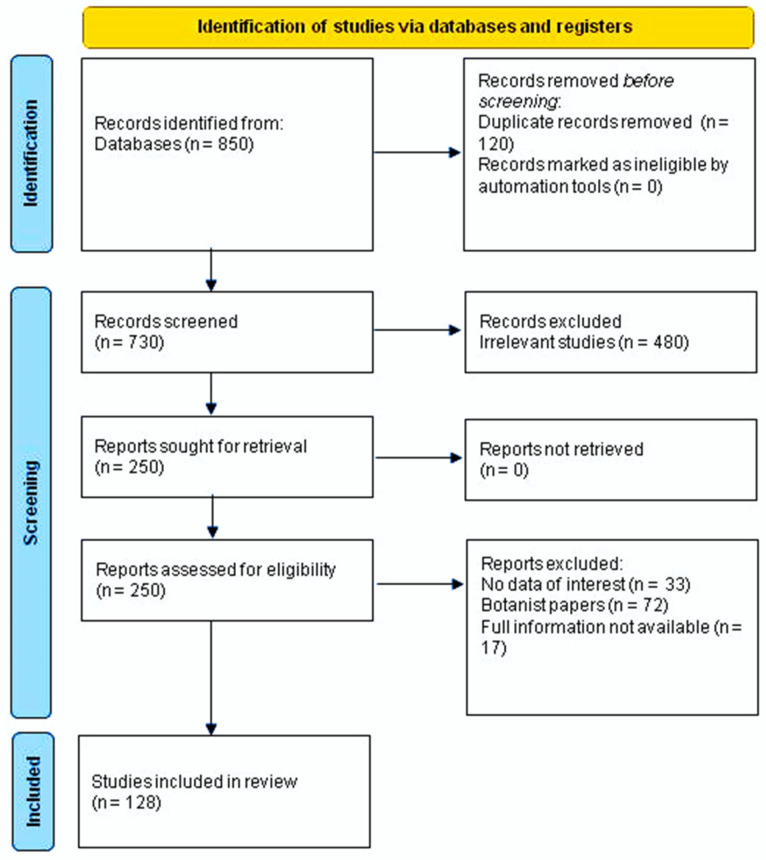
PRISMA 2020 flow diagram for new systematic reviews, which includes database and register searches only.

**Figure 3 plants-11-00519-f003:**
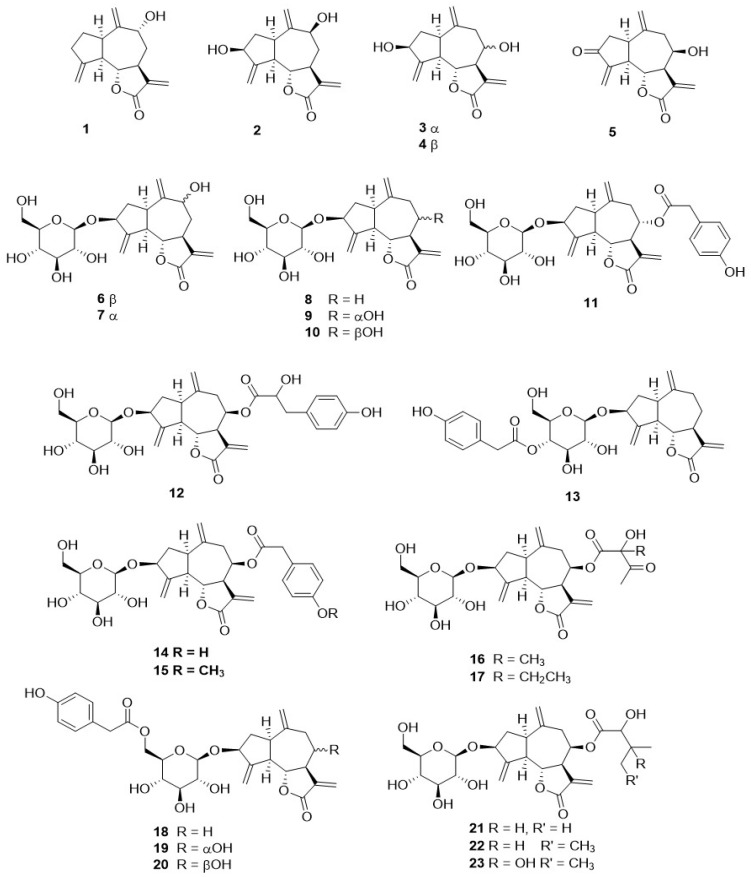
4,15,10,14,11,13-dehydro guaianolides from *Crepis* taxa.

**Figure 4 plants-11-00519-f004:**
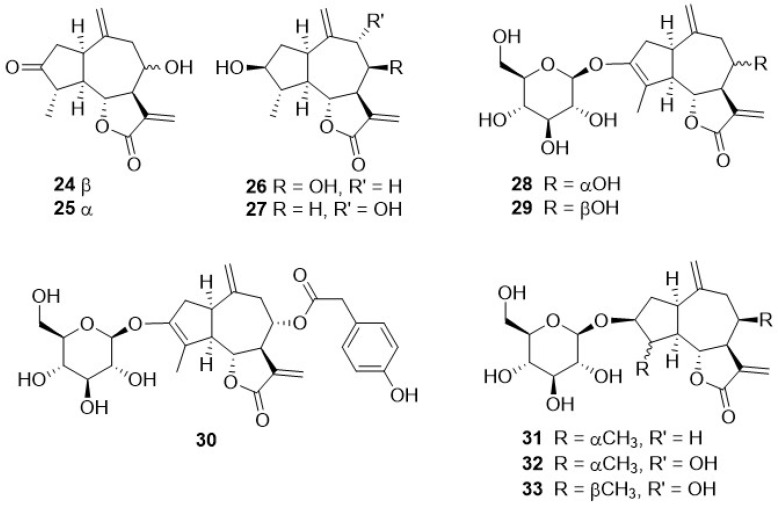
10,14,11,13-dehydro guaianolides from *Crepis* taxa.

**Figure 5 plants-11-00519-f005:**
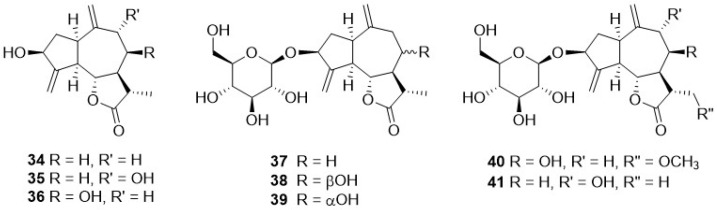
4,15,10,14-dehydro guaianolides from *Crepis* taxa.

**Figure 6 plants-11-00519-f006:**
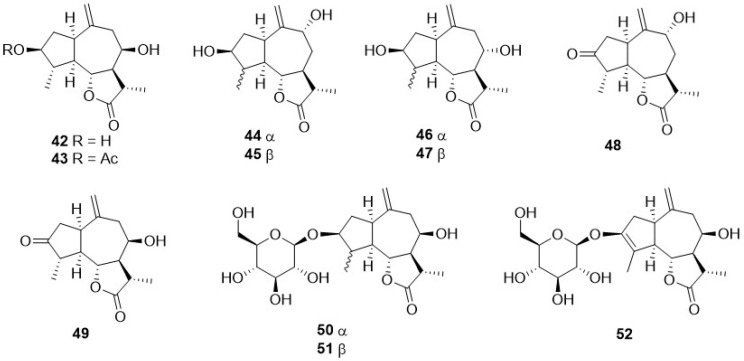
10,14-dehydro guaianolides from *Crepis* taxa.

**Figure 7 plants-11-00519-f007:**
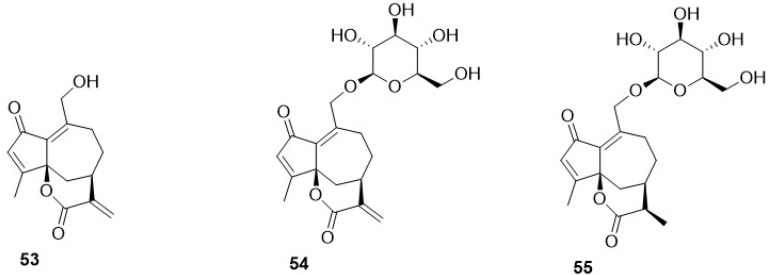
Hypocretenolides from *Crepis* taxa.

**Figure 8 plants-11-00519-f008:**
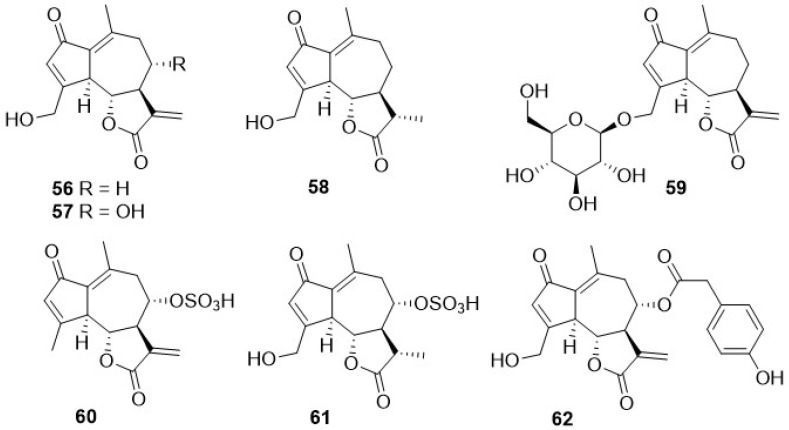
Lactucin type guaianolides from *Crepis* taxa.

**Figure 9 plants-11-00519-f009:**
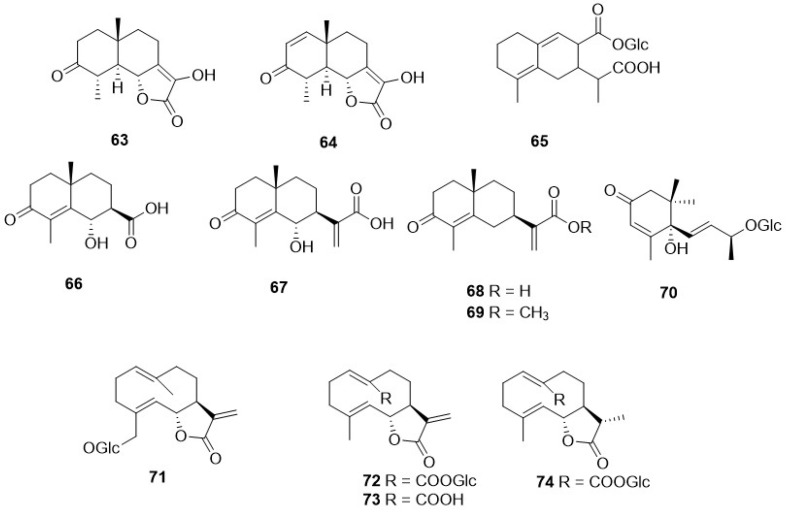
Other sesquiterpenoids from *Crepis* taxa.

**Figure 10 plants-11-00519-f010:**
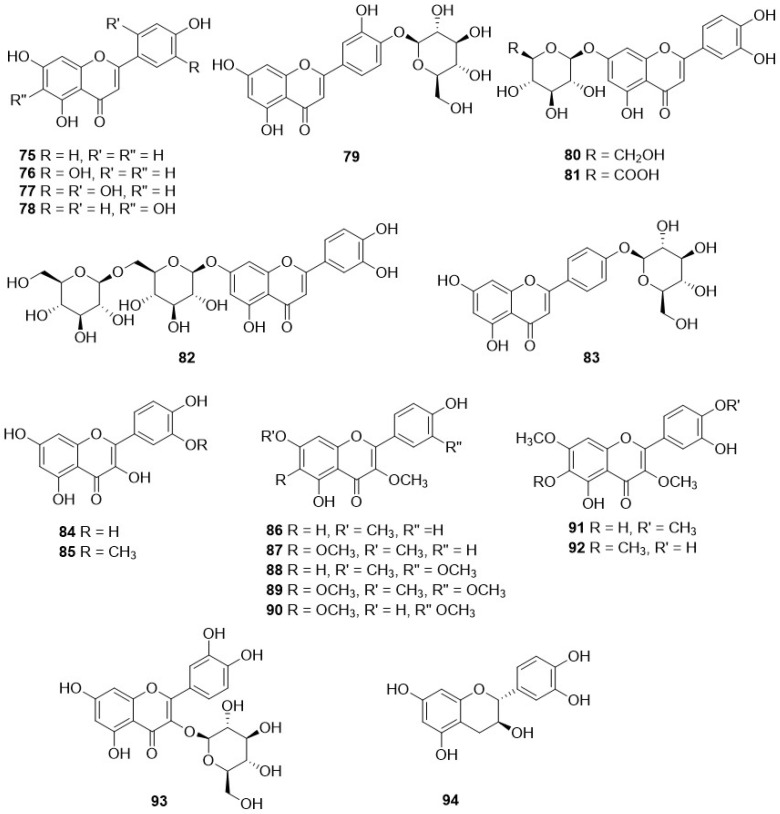
Flavonoids from *Crepis* taxa.

**Figure 11 plants-11-00519-f011:**
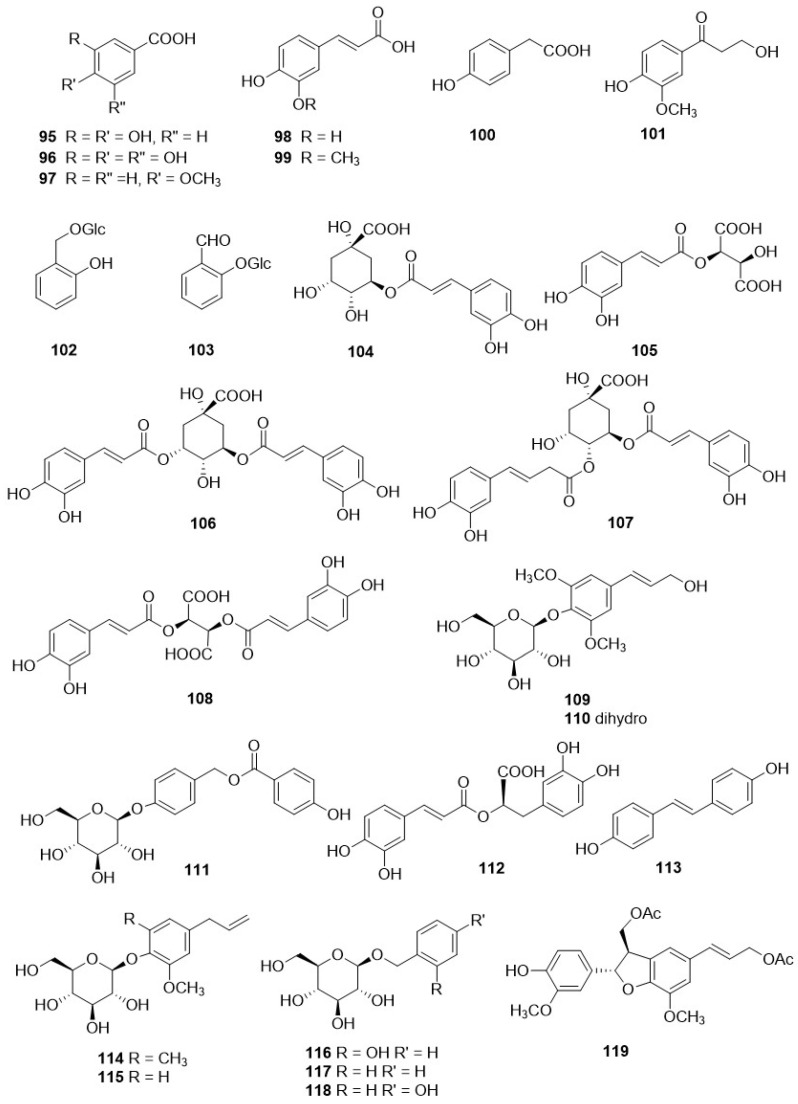
Other metabolites from *Crepis* taxa.

**Figure 12 plants-11-00519-f012:**
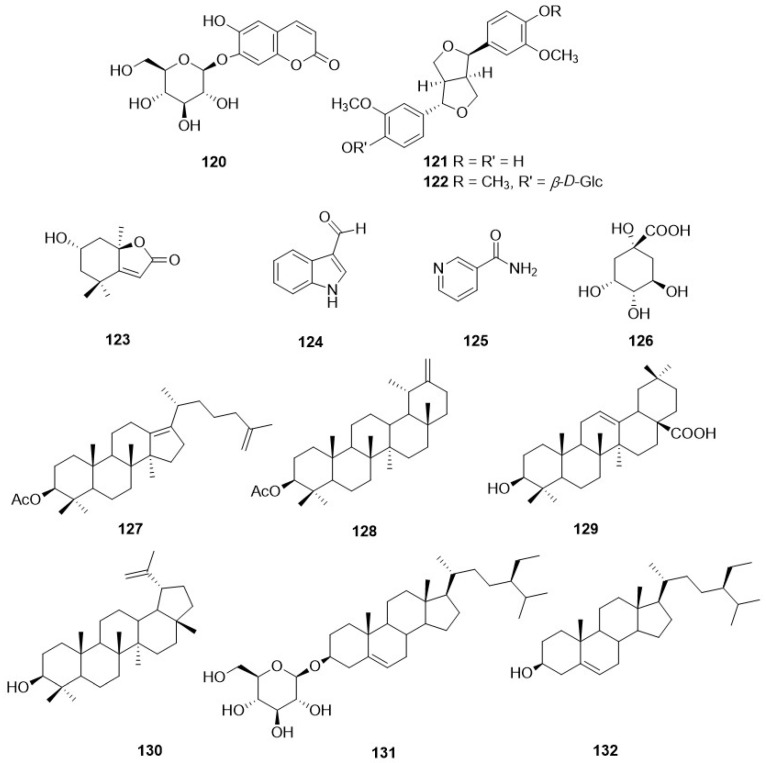
Other metabolites from *Crepis* taxa.

**Table 1 plants-11-00519-t001:** Ethnobotanical use of *Crepis* taxa.

Taxa	Area	Parts	Vernacular Name	Properties	Ref.
*C. biennis*	Abruzzo, Italy	young leaves	cicorietta selvatica	Ingredients of the soup ‘*brodo del pastore*’. Eaten in salads	[26]
*C. bivoniana*	Sicily, Italy	leaves	*luciazzi*	Used in soups, salads and omelettes	[27]
*C. bursifolia*	Campania, Italy	aerial parts	*margarita*	Aerial part used in the preparation of ‘*minestra terrana*’. Basal leaves in salads and soups	[28]
	Sicily, Italy	leaves	*ricuttedda; rizzaredda*	Boiled or pan fried leaves as food. Diuretic and antioxidant	[27,29]
*C. cameroonica*	Cameroon			Treatment of diarrhoea, wounds and fungal infections	[30]
*C. capillaris*	Tuscany, Italy	leaves		Raw or boiled leaves are used for salads and soups. Laxative properties.	[31]
*C. carbonaria*	Africa			Myometrial contractions	[32]
*C. commutata*	Crete, Greece	leaves	*glucosides*	Leaves are usually eaten boiled in salads	[33,34]
*C. crocea*	China			Relieving cough and asthma, antipyretic, benefitting Qi, and reducing inflammation	[35]
*C. flexuosa*	Himalaya	whole plant	*homa-sili*	For jaundice and liver disorder	[36,37]
*C. foetida* ssp. *rhoeadifolia*	Turkey	aerial parts	*yürek out*	For cardiovascular diseases	[38,39]
	Turkey	leaves	*kohum*	Food	[40]
				Anti-cancerous and wound healing agent	[41]
*C. glauca*	Utah, USA	leaves		Leaves have been eaten among Gosiute Indians	[42]
*C. hakkarica*	Turkey	aerial parts	*tahliş*	A decoction of the leaves and flowers is used in the treatment of diabetes	[43]
*C. hyemalis*	Sicily, Italy	leaves	*luciazzi*	Used in soups, salads and with eggs	
*C. japonica*	China			For reducing pyrexia, detoxification and atopy. Antitussive, febrifuge. It is also used in the treatment of boils and snakebites. Internally for the remedies of cold, sore throat, and diarrhoea, and externally as medicated paste to relieve shingles	[44,45]
	Bangladesh	leaves r.	*crepis*	The leaves applied to wounds act as a styptic and heal them quickly. The juice of the root possesses antilithic properties.	[46]
*C. lacera*	Italy			Fatal if frequently ingested by ruminants such as sheep and cattle. Detoxification, purification, diuretic and hypoglycemic effects	[47]
	Campania, Italy	aerial parts		Young basal leaves, flower buds in salads or cooked vegetables	[28]
*C. leontodontoides*	Campania, Italy	aerial parts		Aerial part as ingredient of ‘*minestra terrana*’. Young basal leaves, flower buds in salads or cooked vegetables	[28]
	Tuscany, Italy	leaves		Boiled leaves for salads and soups. Laxative properties	[31]
*C. neglecta*	Tuscany, Italy	leaves		Boiled leaves for salads and soups	[31]
	Campania, Italy	aerial parts	*spaccapreti*	Young basal leaves, flower buds In salads or cooked vegetables.	[28]
*C. neglecta* ssp*. corymbosa*	Campania, Italy	aerial parts	*lattarulo*	Basal leaves in soups	[28]
*C. pulchra*	Campania, Italy	aerial parts		Young basal leaves, flower buds in salads or cooked vegetable	[28]
*C. reuteriana*	Syria	young leaves	*souraga*	As salad for relieve joint diseases pain and as appetiser	[48]
*C. rueppellii*	Yemen			For hepatic disorders (jaundice, hepatitis and gallstones)	[49]
	Ethiopia	roots		Fresh roots are crushed and orally given with water to livestock against Anthrax	[50]
*C. sancta*	Turkey	stems	*düğmelik*	Cooked as digestive	[51]
	Turkey	fleaves	*keklik otu*	For eye diseases and as vasodilators	[52]
	Emilia Romagna (Italy)	young leaves	*ciocapiat*	Food, pan-fried or salad, and for its diuretic and laxative effects	[53]
	Tuscany, Italy	leaves		Boiled leaves are used for salads and soups	[31]
	Umbria, Italy	leaves	*crepide*, *dolcetta*, *radicchiella di terrasanta*	The young leaves are eaten, boiled or raw	[54]
*C. setosa*	Campania, Italy	aerial parts	*occhi neureddi*; *occhi rossi*; *ragno purcello*	Young basal leaves, flower buds in salads or cooked vegetable. Basal leaves eaten in soups or as a cooked vegetable	[28]
	Tuscany, Italy	leaves		Boiled leaves are used for salads and soups	[31]
*C. syriaca*	Syria	young leaves	*souraga*	As salad for relieve joint diseases pain and as appetiser	[48]
*C. vesicaria*	Crete, Greece	leaves		Leaves are usually eaten boiled in salads	[34]
	Abruzzo, Italy	young leaves	*cicorietta selvatica*; *cicorietta di montagna*	Ingredients of soup “brodo del pastore”. Eaten in salads	[26]
	Apulia, Italy	young leaves	*cequariùnné*	Cooked and consumed as vegetable	[55]
	Campania, Italy	aerial parts	*lattarole; cicurioni*; *scazzuoppoli*; *occhi rossi*	Young basal leaves, flower buds in salads or cooked vegetables. Aerial part as ingredient of “*minestra maritata*”. Basal leaves eaten in soups or as a cooked vegetable	[28]
	Tuscany, Italy	leaves		Boiled leaves are used for salads and soups	[31]
	Umbria, Italy	leaves	*crepide*; *dolcetta*; *radicchiella di terrasanta*	The young leaves are eaten, boiled or raw	[54]
	Lucania, Italy	leaves	*marolje*; *marosk*; *liakra sperte*	Among the Arbëresh (ethnic Albanian) communities the young whorls are eaten boiled and fried	[56]
	Sardinia, Italy	leaves	*cicoria areste*	The broth of the boiled leaves is used for abdominal colic. The raw or cooked leaves have hypoglycemic, laxative and hypertensive effects	[57,58]
	Sicily, Italy	leaves	*pizzocorvu*	Boiled leaves with laxative and diuretic effects. Used in soups, salads and with eggs.	[27,29]
*C. vesicaria* ssp. *haenseleri*	Spain	aerial parts	*arnica*; *flor de arnica*	For stomach ailments and problems of arterial circulation; externally applied for wound healing, bruises and inflammations	[59]
*C. vesicaria* ssp. *taraxicifolia*	Emilia Romagna (Italy)	young leaves	*radicchiella*; *strecapugno*	Salads, pan-fried, omelettes, pasta dough. Depurative, refreshing, blood cleaning, diuretic, laxative (cooking water, food)	[53]

**Table 2 plants-11-00519-t002:** Sesquiterpenes from *Crepis* ssp.

Taxa	Origin	Parts	Sesquiterenes	Ref.
*C. aspera*	Egypt	aerial parts	8-*epi*-isolipidiol (**42**)	[64]
*C. aurea*	Poland	roots	14-hydroxy-hypocretenolide (**53**), 14-hydroxy-hypocretenolide-*β*-*D*-glucopyranoside (**54**), 11,13*α*-dihydro-14-hydroxy-hypocretenolide-*β*-*D*-glucopyranoside (**55**)	[65]
	Poland	roots	8-*epi*-deacylcynaropicrin (**4**), integrifolin-3-*O*-*β*-*D*-glucopyranoside (**10**), ixerisoside A (**14**), **15**, ixerin N (**21**), ixerin M (**22**), 8*β*-hydroxy-4*β*,15-dihydrozaluzanin C (**26**), **40**, 8-*epi*-isolipidiol (**42**), 8-*epi*-isolipidiol-3-*O*-*β*-*D*-glucopyranoside (**50**)	[66]
*C. biennis*	Poland	roots	ixerin F (**41**)	[67]
*C. cameroonica*	Cameroon	aerial parts	3*β*,9*β*-dihydroxyguaian-4(15),10(14),11(13)-trien-6,12-olide (**2**), 8-desacylcynaropicrin (**3**), 8*α*-hydroxy-4*α*(15),11*β*(13)-tetrahydrozaluzanin C (**47**)	[30]
*C. capillaris*	Poland	roots	integrifolin (8-*epi*-desacylcynaropicrin) (**4**), integrifolin-3-*O*-*β*-*D*-glucopyranoside (**10**), **16**, **17**, 8-*epi*-isolipidiol (**42**), 8-*epi*-isolipidiol-3-*O*-*β*-*D*-glucopyranoside (**50**)	[68]
	Poland	aerial parts	integrifolin (8-*epi*-desacylcynaropicrin) (**4**), 8-*epi*-isolipidiol (**42**), 8-*ep*i-isoamberboin (**49**)	[69,70]
*C. commutata*	Greece	aerial parts	integrifolin-3-*O*-*β*-*D*-glucopyranoside (**10**), 8-*epi*-grosheimin (**24**), 8-*epi*-isolipidiol (**42**), 3-acetyl-8-*epi*-isolipidiol (**43**), 8-*epi*-isoamberboin (**49**)	[71]
*C. conyzifolia*	Poland	roots	8*β*-hydroxy-4*β*,15-dihydrozaluzanin C (**26**), 4 *β*,15,1l*β*,13-tetrahydrozaluzanin-C-3-*O*-*β*-glucopyranoside (**31**), 8-*epi*-isolipidiol (**42**), 8-*epi*-isolipidiol-3*O*-*β*-*D*-glucopyranoside (**50**)	[72]
*C. crocea*	Mongolia	aerial parts	integrifolin (8-*epi*-desacylcynaropicrin) (**4**), 1l*β*,13-dihydrointegrifolin (**36**), 8-*ep*i-isoamberboin (**49**)	[73]
*C. dioscoridis*	Greece	aerial parts	integrifolin (**4**), integrifolin-3*O*-*β*-*D*-glucopyranoside (**10**), 8-*epi*-grosheimin (**24**), crepiside C (**29**), 1l*β*,13-dihydrointegrifolin (**36**), 1l*β*,13-dihydrointegrifolin-3-*O*-*β*-*D*-glucopyranoside (**38**), 11*β*,13-dihydro-8-*β*-hydroxy-4*α*,15-dihydrozaluzanin 8-*epi*-isolipidiol-3-*O*-*β*-*D*-glucopyranoside (**50**), C-3-*O*-*β*-*D*-glucopyranoside (**51**), 1l*β*,13-dihydrocrepiside C (**52**), 8-deoxylactucin (**56**)	[74]
*C. foetida*	Poland	roots	9α-hydroxy-1l*β*,13-dihydrozaluzanin (**35**), 1l*β*,13-dihydroglucozaluzanin C (**37**), ixerin F (**41**), **44**, **45**	[75]
*C. incana*	Greece	aerial parts	crepiside E (**9**), grosheimin (**25**), crepiside D (**28**), taraxinic acid-1′*O*-*β*-*D*-glucopyranoside (**72**), taraxinic acid (**73**)	[76]
*C. japonica*	Japan	whole plant	glucozaluzanin C (**8**), crepiside E (**9**), crepiside G (**11**), crepiside B (**13**), crepiside A (**18**), crepiside H (**19**), crepiside I (**20**), crepiside D (**28**), crepiside C (**29**), crepiside F (**30**)	[77]
*C. lacera*	Italy	aerial parts	integrifolin-3-*O*-*β*-*D*-glucopyranoside (**10**), 8-*epi*-grosheimin (**24**), 8-*β*-hydroxydehydrozaluzanin C (**5**), 1l*β*,13-dihydrointegrifolin-3-*O*-*β*-*D*-glucopyranoside (**38**)	[47]
*C. leontodontoides*	Italy	aerial parts	8-deoxylactucin (**56**), lactucin (**57**), 1l*β*,13-dihydro-8-deoxylactucin (**58**), 15-deoxylactucin-8-sulfate (**60**), 1l*β*,13-dihydrolactucin-8-sulfate (**61**)	[78]
*C. micrantha*	Egypt	aerial parts	8-desacylcynaropicrin (**3**), 1l*β*,13-dihydrointegrifolin (**36**), 8-*epi*-isolipidiol (**42**)	[79]
*C. mollis*	Poland	roots	integrifolin-3-*O*-*β*-*D*-glucopyranoside (**10**), **16**, **17**, 8-*epi*-grosheimin (**24**), 1l*β*,13-dihydrozaluzanin C (**34**), 1l*β*,13-dihydrointegrifolin-3-*O*-*β*-*D*-glucopyranoside (**38**), **40**, ixerin F (**41**), 8-*epi*-isolipidiol (**42**), 9*α*-hydroxy-4b,15,1l*β*,13-tetra-hydro-dehydrozaluzanin C (**48**), 8-*epi*-isolipidiol-3-*O*-*β*-*D*-glucopyranoside (**50**), picriside B (**71**)	[80]
*C. multicaulis*	Kazakh.	aerial parts	crepidioside A (**59**)	[81]
*C. napifera*	China	roots	napiferoside (**65**)	[82]
	China	roots	taraxinic acid-1′*O*-*β*-*D*-glucopyranoside (**72**), 1l*β*,13-dihydro-taraxinic acid-1′*O*-*β*-*D*-glucopyranoside (**74**)	[83]
*C. pannonica*	Poland	roots	9*α*-hydroxy-3-deoxyzaluzanin C (**1**), macrocliniside A (**7**), integrifolin-3*O*-*β*-*D*-glucopyranoside (**10**), pannonicoside (**23**), 8*β*-hydroxy-4*β*,15-dihydrozaluzanin C-3*O*-*β*-*D*-glucopyranoside (**32**), 8-*β*-hydroxy-4*α*,15-dihydrozaluzanin C-3-*O*-*β*-*D*-glucopyranoside (**33**), **40**, ixerin F (**41**), 8-*epi*-isolipidiol-3*O*-*β*-*D*-glucopyranoside (**50**), 11*β*,13-dihydro-8-*β*-hydroxy-4*α*,15-dihydrozaluzanin C-3-*O*-*β*-*D*-glucopyranoside (**51**)	[84]
*C. pulchra*	Poland	roots	diaspanosise A (**6**), macrocliniside A (**7**), glucozaluzanin C (**8**), 8-*ep*i-isoamberboin (**49**)	[85]
*C. pygmaea*	Italy	whole plant	1,2-4,5-tetrahydro-1l-nor-l1-hydroxy-Δ^7,11^-santonin (**63**), 4,5-dihydro-1l-nor-l1-hydroxy-Δ^7,11^-santonin (**64**)	[86,87]
	Poland	roots	macrocliniside A (**7**), integrifolin-3*O*-*β*-*D*-glucopyranoside (**10**), **16**, **17**, 8*β*-hydroxy-4*β*,15-dihydrozaluzanin C-3*O*-*β*-*D*-glucopyranoside (**32**), ixerin F (**41**), 8-*epi*-isolipidiol-3*O*-*β*-*D*-glucopyranoside (**50**)	[88]
*C. rhoeadifolia*	Poland	roots	1l*β*,13-dihydroglucozaluzanin C (**37**), ixerin F (**41**), **44**, **45**	[89]
*C. sancta*	Jordan	aerial parts	3-oxo-di-nor-eudesm-4-en-6α-hydroxy-11-oic acid (**66**), 3-oxo-6*β*-hydroxy-*γ*-costic acid (**67**), 3-oxo-*γ*-costic acid (**68**), 3-oxo-*γ*-costic acid methyl ester (**69**), (6*R*,9*S*)-roseoside (**70**)	[90]
	Jordan	aerial parts	3-oxo-*γ*-costic acid (**68**), 3-oxo-*γ*-costic acid methyl ester (**69**)	[91]
	Italy	aerial parts	8-deoxylactucin (**56**), 1l*β*,13-dihydro-8-deoxylactucin (**58**), lactucopicrin (**62**)	[78]
*C. setosa*	Poland	roots	9α-hydroxy-1l*β*,13-dihydrozaluzanin (**35**), 1l*β*,13-dihydroglucozaluzanin C (**37**), 8α-hydroxy-1l*β*,13-dihydroglucozaluzanin (**39**), ixerin F (**41**)	[92]
*C. sibirica*	Poland	roots	glucozaluzanin C (**8**), integrifolin-3*O*-*β*-*D*-glucopyranoside (**10**), 8-*ep*i-isoamberboin (**49**)	[93]
*C. tectorum*	Poland	roots	glucozaluzanin C (**8**), tectoroside (**12**), **16**, **17**, ixerin F (**41**), isolipidiol (**46**), 8-*epi*-isolipidiol-3-*O*-*β*-*D*-glucopyranoside (**50**)	[94,95]
	Kazakh.	aerial parts	integrifolin (8-*epi*-desacylcynaropicrin) (**4**), 8-*epi*-isolipidiol (**42**)	[81,96]
*C. tingitana*	Spain	sub aerial parts	integrifolin-3*O*-*β*-*D*-glucopyranoside (**10**), ixerisoside A (**14**)	[97]
*C. vesicaria*	Italy	aerial parts	8-deoxylactucin (**56**), 1l*β*,13-dihydro-8-deoxylactucin (**58**)	[78]
*C. virens*	Italy	flowers	8-*epi*-grosheimin (**24**)	[98]
*C. zacintha*	Poland	roots	integrifolin-3*O*-*β*-*D*-glucopyranoside (**10**), ixerin N (**21**), 8*β*-hydroxy-4*β*,15-dihydrozaluzanin C (**26**), **27**, 8*β*-hydroxy-4*β*,15-dihydrozaluzanin C-3*O*-*β*-*D*-glucopyranoside (**32**), 8-*β*-hydroxy-4*α*,15-dihydrozaluzanin C-3-*O*-*β*-*D*-glucopyranoside (**33**), 1l*β*,13-dihydroglucozaluzanin C (**37**), ixerin F (**41**), **44**, **45**, 8-*epi*-isolipidiol-3-*O*-*β*-*D*-glucopyranoside (**50**), 11*β*,13-dihydro-8-*β*-hydroxy-4*α*,15-dihydrozaluzanin C-3-*O*-*β*-*D*-glucopyranoside (**51**)	[99]

**Table 3 plants-11-00519-t003:** Flavonoids from *Crepis* taxa.

Taxa	Origin	Parts	Sesquiterenes	Ref.
*C. alpestris*	Austria	aerial parts	luteolin (**76**), luteolin 7-*O*-glucoside (**80**), luteolin 7-*O*-glucuronide (**81**), luteolin 7-*O*-gentiobioside (**82**), luteolin 4′-*O*-glucoside (**83**)	[100]
*C. aurea*	Austria	aerial parts	luteolin (**76**), luteolin 7-*O*-glucoside (**80**), luteolin 7-*O*-glucuronide (**81**)	[100]
*C. biennis*	Austria	aerial parts	luteolin (**76**), luteolin 7-*O*-glucoside (**80**), luteolin 7-*O*-glucuronide (**81**), luteolin 7-*O*-gentiobioside (**82**)	[100]
*C. capillaris*	Austria	aerial parts	luteolin (**76**), luteolin 7-*O*-glucoside (**80**),	[100]
	Spain	aerial parts	luteolin (**76**), luteolin 7-*O*-glucoside (**80**), luteolin 4′-*O*-glucoside (**83**)	[101]
	New Zeland	flowers	luteolin (**76**), luteolin 7-*O*-glucoside (**80**), luteolin 7-*O*-glucuronide (**81**)	[102]
*C. commutata*	Greece	aerial parts	luteolin (**76**), luteolin 7-*O*-glucuronide (**81**)	[71]
*C. conyzifolia*	Austria	aerial parts	luteolin (**76**), luteolin 7-*O*-glucoside (**80**), luteolin 7-*O*-glucuronide (**81**), luteolin 4′-*O*-glucoside (**79**)	[100]
*C. dioscoridis*	Greece	aerial parts	apigenin-4′-*O*-glucoside (**83**), isorhamnetin (**85**)	[74]
*C. divaricata*			luteolin (**76**), quercetin (**84**)	[103]
*C. foetida*	Italy	aerial parts	luteolin (**76**), luteolin 7-*O*-glucoside (**80**), luteolin 7-*O*-gentiobioside (**82**)	[100]
*C. foetida* ssp. *rhoeadifolia*	Turkey	flowers	apigenin (**75**), quercetin (**84**), (+)-catechin (**94**)	[104]
	Turkey	aerial parts	luteolin (**76**)	[105]
*C. froelichiana*.	Italy	aerial parts	luteolin (**76**), luteolin 7-*O*-glucoside (**80**), luteolin 7-*O*-glucuronide (**81**), luteolin 4′-*O*-glucoside (**83**)	[100]
*C. incana*	Greece	aerial parts	luteolin (**76**), luteolin 7-*O*-glucoside (**80**), quercetin 3-glucoside (**93**)	[76]
*C. jacquinii* ssp. *kerneri*	Italy	aerial parts	luteolin (**76**), luteolin 7-*O*-glucoside (**80**)	[100]
*C. japonica*			luteolin-7-*O*-glucoside (**80**)	[45,106]
*C. micrantha*	Egypt	aerial parts	apigenin (**75**), luteolin (**76**)	[79]
*C. mollis*	Austria	aerial parts	luteolin (**76**), luteolin 7-*O*-glucoside (**80**), luteolin 7-*O*-glucuronide (**81**), luteolin 7-*O*-gentiobioside (**82**)	[100]
*C. nemausensis*	Italy	aerial parts	luteolin (**76**), luteolin 7-*O*-glucoside (**80**), luteolin 7-*O*-glucuronide (**81**), luteolin 4′-*O*-glucoside (**83**)	[100]
*C. paludosa*	Germany	aerial parts	luteolin (**76**), luteolin 7-*O*-glucoside (**80**), luteolin 7-*O*-glucuronide (**81**), luteolin 4′-*O*-glucoside (**83**)	[100]
*C.* *pygmaea*	Italy	aerial parts	luteolin (**76**), luteolin 7-*O*-glucoside (**80**), luteolin 7-*O*-gentiobioside (**82**), luteolin 7-*O*-glucuronide (**81**)	[100]
			luteolin (**76**), quercetin (**84**)	[103]
*C. rhaetica*	Austria	aerial parts	luteolin (**76**), luteolin 7-*O*-glucoside (**80**), luteolin 7-*O*-glucuronide (**81**)	[100]
*C. sancta*	Jordan	aerial parts	kumatakenin (**86**), penduletin (**87**)	[90]
	Jordan	aerial parts	kumatakenin (**86**), penduletin (**87**), pachypodol (**88**), jaceidin (**89**), chrysosplentin (**90**), casticin (**91**) 3,5,7-tri-*O*-methyl-6-methoxy-kaempferol (**92**)	[91]
*C. senecioides*			luteolin (**76**), isoetin (**77**), scutellarein (**78**)	[107]
*C. tectorum*			isoetin (**77**), scutellarein (**78**)	[107]
*C. terglouensis*	Switzerland	aerial parts	luteolin (**76**), luteolin 7-*O*-glucoside (**80**), luteolin 7-*O*-glucuronide (**81**), luteolin 7-*O*-gentiobioside (**82**)	[100]
*C. tingitana*	Spain	aerial parts	luteolin (**76**), luteolin 7-*O*-glucoside (**80**), luteolin 7-*O*-glucuronide (**81**), luteolin 4′-*O*-glucoside (**83**)	[100]
*C. vesicaria*	Spain	aerial parts	luteolin (**76**), luteolin 7-*O*-glucoside (**80**), luteolin 4′-*O*-glucoside (**83**)	[101,108]

**Table 4 plants-11-00519-t004:** Other metabolites present in *Crepis* taxa.

Taxa	Origin	Parts	Compounds	Ref.
*C. alpestris*	Austria	aerial parts	chlorogenic acid (**104**), caffeoyltartaric acid (**105**), 3,5-di-*O*-dicaffeoylquinic acid (1**06**), cichoric acid (**108**)	[100]
*C. aspera*	Egypt	aerial parts	oleanolic acid (**129**), sitosterol 3-*O*-*β*-*D*-glucopyranoside (**131**)	[64]
*C. aurea*	Poland	roots	syringing (**109**), 5-methoxy-eugenyl-4-*O*-*β*-glucopyranoside (**114**)	[66]
	Austria	aerial parts	chlorogenic acid (**104**), caffeoyltartaric acid (**105**), 3,5-di-*O*-dicaffeoylquinic acid (**106**), cichoric acid (**108**)	[100]
*C. biennis*	Austria	aerial parts	chlorogenic acid (**104**), caffeoyltartaric acid (**105**), 3,5-di-*O*-dicaffeoylquinic acid (**106**), cichoric acid (**108**)	[100]
*C. bocconi*	Germany	sub aerial parts	4-hydroxybenzoic acid 4-*β*-D-glucopyranosyloxy-benzyl ester (**111**)	[111]
*C. capillaris*	Austria	aerial parts	chlorogenic acid (**104**), caffeoyltartaric acid (**105**), 3,5-di-*O*-dicaffeoylquinic acid (**106**), cichoric acid (**108**)	[100]
	Spain	aerial parts	chlorogenic acid (**104**), 3,5-di-*O*-dicaffeoylquinic acid (**106**) cichoriin (**120**)	[101]
	New Zeland	flowers	caffeic acid (**98**), chlorogenic acid (**104**), caffeoyltartaric acid (**105**), 3,5-di-*O*-caffeoylquinic acid (**106**), cichoric acid (**108**)	[102]
*C. commutata*	Greece	aerial parts	*p*-anisic acid (**97**), *E*-caffeic acid (**98**), *p*-hydroxyphenylacetic acid (**100**)	[71]
*C. conyzifolia*	Poland	roots	5-methoxy-eugenyl-4-*O*-*β*-glucopyranoside (**114**), eugenyl-4-*O*-*β*-glucopyranoside (**115**), cichoriin (**120**)	[72]
	Austria	aerial parts	chlorogenic acid (**104**), 3,5-di-*O*-dicaffeoylquinic acid (**106**)	[100]
*C. crocea*			polysaccarhides	[35]
*C. dioscoridis*	Greece	aerial parts	*E*-caffeic acid (**98**), ferulic acid (**99**), isosalicin (**116**)	[74]
*C. foetida*	Italy	aerial parts	chlorogenic acid (**104**), caffeoyltartaric acid (**105**), 3,5-di-*O*-dicaffeoylquinic acid (**106**), cichoric acid (**108**)	[100]
*C. foetida* ssp. *rhoeadifolia*	Turkey	floers	protocatechuic acid (**95**), gallic acid (**96**), caffeic acid (**98**), chlorogenic acid (**104**), rosmarinic acid (**112**)	[104]
	Turkey	aerial parts.	chlorogenic acid (**104**)	[105]
	Turkey	roots	chlorogenic acid (**104**)	[105]
*C. froelichiana*	Italy	aerial parts	chlorogenic acid (**104**), caffeoyltartaric acid (**105**), 3,5-di-*O*-dicaffeoylquinic acid (**106**), cichoric acid (**108**)	[100]
*C. incana*	Greece	aerial parts	(3*S*,5*R*)-loliolide (**123**)	[76]
	Greece	roots	oleanolic acid (**129**), lupeol (**130**)	[76]
*C. jacquinii* ssp. *kerneri*	Italy	aerial parts	chlorogenic acid (**104**), caffeoyltartaric acid (**105**), 3,5-di-*O*-dicaffeoylquinic acid (**106**), cichoric acid (**108**)	[100]
*C. japonica*			3,5-di-*O*-dicaffeoylquinic acid (**106**), 3,4-di-*O*-caffeoylquinic acid (**107**)	[45,106]
*C. lacera*	Italy	aerial parts	euphorbol acetate (**127**)	[112]
	Italy	aerial parts	*p*-hydroxy-benzyl 7-*O*-*β*-glucopyranoside (**118**), pynoresynol (**121**)	[47]
*C. mollis*	Poland	roots	3-hydroxy-1-(4-hydroxy-3-methoxyphenyl)-1-propanone (**101**), 5-methoxy-eugenyl-4-*O*-*β*-glucopyranoside (**114**)	[80]
	Austria	aerial parts	chlorogenic acid (**104**), caffeoyltartaric acid (**105**), 3,5-di-*O*-dicaffeoylquinic acid (**106**), cichoric acid (**108**)	[100]
*C. napifera*	China	roots	acetate taraxasterol (**128**)	[82]
*C. nemausensis*	Italy	aerial parts	chlorogenic acid (**104**), caffeoyltartaric acid (**105**), 3,5-di-*O*-dicaffeoylquinic acid (**106**), cichoric acid (**108**)	[100]
*C. paludosa*	Germany	aerial parts	chlorogenic acid (**104**), caffeoyltartaric acid (**105**), 3,5-di-*O*-dicaffeoylquinic acid (**106**), cichoric acid (**108**)	[100]
*C. pannonica*	Poland	roots	syringing (**109**), dihydrosyringin (**110**), 5-methoxy-eugenyl-4-*O*-*β*-glucopyranoside (**114**), eugenyl-4-*O*-*β*-glucopyranoside (**115**), benzyl-*O*-*β*-glucopyranoside (**117**), 3-indolecarbaldehyde (**124**), nicotinamide (**125**)	[84]
*C.* *pygmaea*	Italy	aerial parts	chlorogenic acid (**104**), caffeoyltartaric acid (**105**), 3,5-di-*O*-dicaffeoylquinic acid (**106**), cichoric acid (**108**)	[100]
*C. rhaetica*	Austria	aerial parts	chlorogenic acid (**104**), 3,5-dicaffeoylquinic acid, caffeoyltartaric acid (**105**), cichoric acid (**108**)	[100]
*C. rhoeadifolia*	Poland	roots	isosalicn (**102**), helicin(**103**), dehydrodiconyferyl alcohol diacetate (**119**)	[41]
*C. terglouensis*	Switzerland	aerial parts	chlorogenic acid (**104**), caffeoyltartaric acid (**105**), 3,5-di-*O*-dicaffeoylquinic acid (**106**), cichoric acid (**108**)	[100]
*C. tingitana*	Spain	aerial parts	chlorogenic acid (**104**), 3,5-di-*O*-dicaffeoylquinic acid (**106**)	[100]
*C. turczaniowii*			polysaccarhides	[113]
			phillyrin (**122**), acetate taraxasterol (**128**), *β*-sitosterol (II) (**132**)	[114]
*C. vesicaria*	Spain	aerial parts	chlorogenic acid (**104**), 4,4′-dihydroxy-stilbene (**113**)	[101,108]
*C. vesicaria* ssp. *taraxicifolia*	Portugal	aerial parts	caffeic acid (**98**), cichoric acid (**108**), quinic acid (**126**)	[61]

**Table 5 plants-11-00519-t005:** Fixed oil composition of *Crepis* taxa.

Taxa	Origin	Parts	Fatty Acids	Ref.
*C. alpina*	USA	seeds	crepenynic acid; palmitic acid, stearic acid, oleic acid, linoleic acid	[115]
*C. aurea*	Yugoslavia	seeds	vernolic acid, crepenynic acid, palmitic acid, stearic acid, oleic acid, linoleic acid	[116]
*C. biennis*	Poland	seeds	vernolic acid, crepenynic acid, palmitic acid, stearic acid, oleic acid, linoleic acid	[116]
*C. conyzaefolia*	USA	seeds	(-)-*S*,*S*,-l2-hydroxy-l3-octadec-*cis*-9-enolide, *cis*-12,13-epoxy-octadeca-*trans*-6-cis-9-dienoic acid, *cis*-12,13-epoxyoctadeca-*cis*-6-*cis*-9-dienoic acid, vernolic acid	[117,118]
*C. foetida*	Turkey	seeds	crepenynic acid, palmitic acid, stearic acid, oleic acid, linoleic acid	[116]
*Crepis foetida* ssp. *rhoeadifolia*	Turkey	seeds	crepenynic acid, palmitic acid, stearic acid, oleic acid, linoleic acid	[116]
*C. intermedia*	USA	seeds	vernolic acid, crepenynic acid, palmitic acid, stearic acid, oleic acid, linoleic acid	[116]
*C. lacera*	Italy	aerial parts	palmitoleic acid, stearic acid, palmitic acid, linolenic acid, linoleic acid, oleic acid, isopalmitic acid	[111]
*C. occidentalis*	USA	seeds	vernolic acid, crepenynic acid, palmitic acid, stearic acid, oleic acid, linoleic acid	[116]
*C. rubra*	USA	seeds	crepenynic acid, palmitic acid, stearic acid, oleic acid, linoleic acid	[116]
*C. thomsonii*	Pakistan	seeds	crepenynic acid, palmitic acid, stearic acid, oleic acid, linoleic acid	[116]
*C. vesicaria* ssp. *taraxicifolia*	Spain	seeds	vernolic acid, crepenynic acid, palmitic acid, stearic acid, oleic acid, linoleic acid	[116]
	Portugal	leaves	α-linolenic acid, linoleic acid, oleic acid; palmitic acid, gondoic acid, arachidic acid, stearic acid; margaric acid	[114]

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
