# Peer review of "Ethnobotany, Phytochemistry, Biological, and Nutritional Properties of Genus *Crepis*—A Review"

_plants, 2022, doi:10.3390/plants11040519_

Round 1
Reviewer 1 Report
Dear Authors,
The Manuscript ID: plants-1592608, Titled “The ethnobotany, phytochemistry, biological, and alimurgic properties of genus Crepis – A Review” is well-designed. It presents the purpose of the review, the studies and results on the taxa, and main conclusions.
Based on the evaluation of its originality, significance of content, scientific soundness, and interest to readers, a minor revision is suggested before the review may be considered for acceptance. Specific suggestions and comments are provided below.
The title is concise and informative. It declares the object of the review.
The Abstract is factual and well-structured. It states briefly the phytochemical investigations on genus Crepis, the main secondary metabolites and the used extraction methods, its ethnomedicinal and pharmacological uses. Though, the main analytical methods for identification and elucidation of the compounds should be written too.
A critical aspect is announced: the necessity for extensive investigations on the real capacity of different Crepis species used in food, and to analyze promising new species on a scientific level.
The Introduction presents detailed data concerning the studied genus. The literature survey corresponds to the object of the review.
The systematical, ethnobotanical data and the phytochemical constituents (sesquiterpenes and flavonoids and others) of the studied plant genus as well as the biological and alimurgic properties of Crepis spp. are presented in details. However, I should suggest a subheading with a review on the methods used for identification, tentatively elucidation or confirmation of the chemical constituents.
Data on chemotaxonomic significance of Crepis taxa are announced too.
The Conclusions are well summarized. The critical aspects are declared considering the further investigations on non-studied species.
Author Response
Rewiew 1
Dear Authors,
The Manuscript ID: plants-1592608, Titled “The ethnobotany, phytochemistry, biological, and alimurgic properties of genus Crepis – A Review” is well-designed. It presents the purpose of the review, the studies and results on the taxa, and main conclusions.
Based on the evaluation of its originality, significance of content, scientific soundness, and interest to readers, a minor revision is suggested before the review may be considered for acceptance. Specific suggestions and comments are provided below.
The title is concise and informative. It declares the object of the review.
The Abstract is factual and well-structured. It states briefly the phytochemical investigations on genus Crepis, the main secondary metabolites and the used extraction methods, its ethnomedicinal and pharmacological uses. Though, the main analytical methods for identification and elucidation of the compounds should be written too.
The main analytical methods have been inserted
A critical aspect is announced: the necessity for extensive investigations on the real capacity of different Crepis species used in food, and to analyze promising new species on a scientific level.
The Introduction presents detailed data concerning the studied genus. The literature survey corresponds to the object of the review.
The systematical, ethnobotanical data and the phytochemical constituents (sesquiterpenes and flavonoids and others) of the studied plant genus as well as the biological and alimurgic properties of Crepis spp. are presented in details. However, I should suggest a subheading with a review on the methods used for identification, tentatively elucidation or confirmation of the chemical constituents.
A paragraph has been inserted in section 4.1.
Data on chemotaxonomic significance of Crepis taxa are announced too.
The Conclusions are well summarized. The critical aspects are declared considering the further investigations on non-studied species.
Reviewer 2 Report
The review entitled “The Ethnobotany, Phytochemistry, Biological, and Alimurgic Properties of Genus Crepis – A Review” has been submitted to the special issue “Phytochemical and Nutritional Analysis of Medicinal and Aromatic Plants”.
The manuscript falls in the scope of Plants; however, I’ve some concerns if it falls in the scope of the mentioned SI, which is particularly dedicated to aromatic plants, such herbs and spices.
The manuscript aims at reporting the ethnobotany, phytochemistry, and biological activities of the species from Crepis genus. This is a literature review and the data was collected from research databases covering a very large time span from 1911 to 2021, comprising a total of 134 references)
The manuscript is on average well written and represents a suitable contribution to the subject. However, some revisions and corrections are needed before being accepted for publication.
- The title must be revised. It also refers to the alimurgic properties of Crepis genus that were not described in the main text.
- This is a literature review, it´s not a systematic review. Corrections are needed in lines 21 and 558.
- Line 17: Do you mean ethnobotanical level?
- Lines 24 – 27: nothing is described in the main text regarding the different extraction procedures.
- The authors must clearly explain why do they choose Crepis genus to do a literature review. What kind of articles are already published. Is there some review on the subject or is this the first one? What is the main contribution of this review?
- Lines 90-95: what were the criteria used to exclude publications from the review?
- Some English corrections are needed: line 109 (locally), line 197 (sesquiterpenes), Line 241 (clearly increase), line 242 (altitude), line 254 (represented), lines 289 and 296 (on the other hand); an English revision is advisable.
- Regarding the description of biological activities, only a few studies are reported. A real critical assessment is mandatory because most of the studies described concentrations of plant extracts and pure compounds at non-significant levels. It is advisable to read: Cos P, Vlietinck AJ, Berghe D Vanden, Maes L (2006) Anti-infective potential of natural products: How to develop a stronger in vitro “proof-of-concept.” J Ethnopharmacol 106:290–302. doi: 10.1016/j.jep.2006.04.003, and Gertsch J, How scientific is the science in ethnopharmacology? Historical perspectives and epistemological problems. Ethnopharmacology, 2009, 122: 177-183, in order to improve the discussion.
- Lines 317-318: Please clarify this sentence.
- The antioxidant results described are all obtained using chemical assays; this subsection is not well included in the biological properties section.
- Line 397: HPLC-DAD: high-performance chromatography with a diode array detector. Please correct.
- Line 477: what do you mean by “Tossicity”? Is it toxicity? The authors should revise this part of the text; Why did you divide sections 6.7 and 6.8? They are all about toxicity and cytotoxicity.
- Section 6.9. Nutritional composition. This subsection is not well located here because it´s not about biological properties. It should be located after the traditional uses.
- Change and improve Figure 12 in accordance with my previous suggestions.
- Lines 550-552: This review describes only a few studies on the biological activities of Crepis species. They are not enough to validate or confirm the ethnomedicinal or edible uses of the plants. Therefore, this kind of sentence is somewhat very speculative and should be avoided. Please revise.
- The conclusion must be revised and improved: the paper as it is, seems only a collection of data and lacks a proper discussion, particularly regarding the biological properties. It´s mandatory to make a critical assessment of all the collected data. An improved discussion and conclusion will help the reader to make decisions about future studies directions, and what is already shown to be promising and not promising. Some clear conclusions for future work should also be given.
Author Response
Review 2
- The title must be revised. It also refers to the alimurgic properties of Crepis genus that were not described in the main text.
Title has been modified
- This is a literature review, it´s not a systematic review. Corrections are needed in lines 21 and 558.
Corrected
- Line 17: Do you mean ethnobotanical level?
Modified
- Lines 24 – 27: nothing is described in the main text regarding the different extraction procedures.
Inserted in paragraph 4.1.
- The authors must clearly explain why do they choose Crepis genus to do a literature review. What kind of articles are already published. Is there some review on the subject or is this the first one? What is the main contribution of this review?
A phrase has been inserted with the explanations and same references have been added.
- Lines 90-95: what were the criteria used to exclude publications from the review?
Articles used have been chosen basing on principles of PRISMA (Figure 2)
- Some English corrections are needed: line 109 (locally), line 197 (sesquiterpenes), Line 241 (clearly increase), line 242 (altitude), line 254 (represented), lines 289 and 296 (on the other hand); an English revision is advisable
Corrected
- Regarding the description of biological activities, only a few studies are reported. A real critical assessment is mandatory because most of the studies described concentrations of plant extracts and pure compounds at non-significant levels. It is advisable to read: Cos P, Vlietinck AJ, Berghe D Vanden, Maes L (2006) Anti-infective potential of natural products: How to develop a stronger in vitro “proof-of-concept.” J Ethnopharmacol 106:290–302. doi: 10.1016/j.jep.2006.04.003, and Gertsch J, How scientific is the science in ethnopharmacology? Historical perspectives and epistemological problems. Ethnopharmacology, 2009, 122: 177-183, in order to improve the discussion.
Thank you for your suggestion. In this literature review the aim was to describe all the available data on this genus rather than a comparison with other genus
- Lines 317-318: Please clarify this sentence.
Revised
- The antioxidant results described are all obtained using chemical assays; this subsection is not well included in the biological properties section.
Thank for suggestions. We reported in this section all the available data of literature on this topic. Of course the absence of other assays besides of the chemistry methods is a criticism, that has been now highlighted in the text.
- Line 397: HPLC-DAD: high-performance chromatography with a diode array detector. Please correct.
Corrected
- Line 477: what do you mean by “Tossicity”? Is it toxicity? The authors should revise this part of the text; Why did you divide sections 6.7 and 6.8? They are all about toxicity and cytotoxicity.
Toxicity corrected. The section 6.7 focuses only the toxicity against particular cell lines. The next one reports, two works on hepatotoxicity in vivo on mouse and alimentary toxicity on cattle.
- Section 6.9. Nutritional composition. This subsection is not well located here because it´s not about biological properties. It should be located after the traditional uses.
Section 6.9 has been moved to Section 3.2
- Change and improve Figure 12 in accordance with my previous suggestions
Figure removed
- Lines 550-552: This review describes only a few studies on the biological activities of Crepis species. They are not enough to validate or confirm the ethnomedicinal or edible uses of the plants. Therefore, this kind of sentence is somewhat very speculative and should be avoided. Please revise.
A new sentence has been introduced in the conclusion.
- The conclusion must be revised and improved: the paper as it is, seems only a collection of data and lacks a proper discussion, particularly regarding the biological properties. It´s mandatory to make a critical assessment of all the collected data. An improved discussion and conclusion will help the reader to make decisions about future studies directions, and what is already shown to be promising and not promising. Some clear conclusions for future work should also be given.
Conclusion were modified. The critical aspect was already reported, highlighting he lake of in-depth biological studies.
Reviewer 3 Report
This is an apparently useful review since it provides a thorough dataset on the various properties of genus Crepis. The review is generally well written and structured, and provides a balanced view of the topic. I suggest acceptance of this review article in its present form.
Author Response
Thank you for your comments
Reviewer 4 Report
The manuscript entitled “The Ethnobotany, Phytochemistry, Biological, and Alimurgic Properties of Genus Crepis – A Review” provides comprehensive information of traditional uses, phytochemical compositions, and biological properties of the genus Crepis.
The review was quite well performed, however, to be accepted by the Plants journal, authors need to consider some issues as follows:
- The plagiarism checking software (iThenticate) shows 28% similarity. Authors must re-write the following parts: Title, Line 84-97, Line 157-160.
- English grammar and expression should be checked carefully throughout the manuscript. E.g. Line 28: “have been and are used as” should be more simple as “have been used as”. Please carefully check every sentence.
- Figure 1: The indicator letters should be presented more clearly. (E) should be vertical. The sub-pictures must be cited from public sources or copyright sources.
- Review methodology: should be presented based on the principles of PRISMA.
- Table 1 and other Tables: The same species should be gathered in one cell. Local names should be spelled in the Latin alphabet. Used plant part column should be presented in full form but not in abbreviations.
- Shorten the number of figures. E.g. authors should show the general structures of major compound groups found in Crepis Also, why did authors separate Fig. 10 and 11?
- Figure 12: the photos from the original studies of authors or cited from other sources? Authors must ensure the copyright of all photos (pictures) used in this review.
- Section 6.8. Tossicity (English mistake): Along with the hepatoprotective effect of some Crepis species, authors should discuss potential hepatotoxicity of some unknown species as well as other phytocompounds involved in the plant parts as the following manuscripts did 10.3390/ijms21145011 and 10.3390/ijms221910419.
- Authors should avoid using the colon “ : ” in writing a scientific paper.
- Caption and performance of Table S1 must be revised.
Author Response
Review 4
- The plagiarism checking software (iThenticate) shows 28% similarity. Authors must re-write the following parts: Title, Line 84-97, Line 157-160.
The parts have been modified
- English grammar and expression should be checked carefully throughout the manuscript. E.g. Line 28: “have been and are used as” should be more simple as “have been used as”. Please carefully check every sentence.
Done
- Figure 1: The indicator letters should be presented more clearly. (E) should be vertical. The sub-pictures must be cited from public sources or copyright sources.
The pictures are personal. and the Figure has been corrected
- Review methodology: should be presented based on the principles of PRISMA.
Inserted
- Table 1 and other Tables: The same species should be gathered in one cell. Local names should be spelled in the Latin alphabet. Used plant part column should be presented in full form but not in abbreviations.
Modified
- Shorten the number of figures. E.g. authors should show the general structures of major compound groups found in Crepis Also, why did authors separate Fig. 10 and 11?
Several structures have been grouped. Fig 10 and 11 have been unified
- Figure 12: the photos from the original studies of authors or cited from other sources? Authors must ensure the copyright of all photos (pictures) used in this review.
Figure 12 deleted
- Section 6.8. Tossicity (English mistake): Along with the hepatoprotective effect of some Crepis species, authors should discuss potential hepatotoxicity of some unknown species as well as other phytocompounds involved in the plant parts as the following manuscripts did 10.3390/ijms21145011 and 10.3390/ijms221910419.
A paraghaph has been inserted
- Authors should avoid using the colon “ : ” in writing a scientific paper.
Corrected
- Caption and performance of Table S1 must be revised.
Corrected
Reviewer 5 Report
This review systematically presents the results of research on the genus Crepis. Different plant extracts and various compounds isolated from Crepis species have been evaluated for their biological activities (antitumor, anti-inflammatory, antiviral, antimicrobial, anti-ulcer, antioxidant, and nutritional). The secondary metabolites within the genus have demonstrated significant medicinal potential. The paper presents updated report of the current literature related to the nutritional properties and biological activities of the genus Crepis and confirms its phytotherapeutic potential.
The abstract of the manuscript is appropriate for the content of the text, the article is well constructed and well interprets the significance of the findings described in the cited literature references.
After minor revision, the paper might be accepted in Plants as a Review.
The following comments, recommendations and corrections should be taken into consideration:
General comments
- It is recommended to change the title as follows: Ethnobotany, Phytochemistry, Biological Activity, and Alimurgic Properties of Genus Crepis – A Review.
- In vitro and in vivo terms should be in italic (should be checked throughout the manuscript).
Specific comments
Line 109: Change locxally into locally.
Line 151: Table 1 (line 4 – C. bursifolia, Sicily): change antoxidant into antioxidant; (line C. sancta, Tuscany): column Properties – correction required.
Line 170: Table 2 (line 3 – C. aurea, Poland): change 8-epi-isolippidiol into 8-epi-isolipidiol.
Line 184: Change researches into research.
Line 194: (Fig.6) – insert space.
Line 241: Change clerly into clearly.
Lines 254 and 255: Wild herbs of the Asteraceae family should be named in Latin.
Line 297: …of the from the… Correction required.
Line 309: Figure 12. Tossicity should be Toxicity.
Line 358: The square bracket is missing: [121].
Lines 358-362: The sentence should be rearranged to be clearer.
Line 386: …, whereas…
Line 388: Remove of (…of eicosanoic acid…of eneicosane).
Line 423: At doses of 100 mg/kg… (should be inserted).
Line 449: Should be 6.7. Cytotoxic Activity or 6.7. Cytotoxicity
Lines 457-460: The sentence should be rearranged to be clearer.
Line 461: Oriental hawksbeard should be named in Latin.
Line 477: Should be 6.8. Toxicity
Line 525: …source of xantophylls… (should be inserted).

Author Response
Review 5
All the suggestions have been accepted and the manuscript has been modified
Round 2
Reviewer 2 Report
The authors revised the manuscript taking into consideration most of my suggestions. It can now be published in Plants. An English spell check is mandatory since I detected some errors.
Author Response
Dear reviewer, thanks for your comments. Several corrections of the English style and of misprints have been done.
Reviewer 4 Report
The authors attempted to correct the manuscript according to my recommendations. And I was very impressed with their very polite responses to the reviewer.
Some more comments are as follows:
English mistakes still exist throughout the manuscript. For example, L85, principkes --> principles; Table S1, caption, synonymous --> synonyms; L489: C5N e A5 --> C5N and A5 cell lines, etc.
Many colons in academic writing are not recommended. Please find other expressions. E.g. abstract section and others. Please do not use speaking language in writing, e.g. just thinking that, of course, etc.
Figure 2. PRISMA Diagram should be presented in higher resolution. What is the meaning of the asterisks "*" in the figure? and Why is 2020?
In summary, I suggest a minor revision to this manuscript. Authors must do an extensive revision focused on English.
The academic value of this manuscript is quite high, so please make it as perfect as possible.
Author Response
Dear review, thank you very much for your comments.
Prisma fig is present now at higher resolution. We used, as per ref 25, the last version of PRISMA diagram.
Several corrections of english style and mistprints have been done.